# VeriRole: Verifiable Role-Awareness through Hint-Guided Reinforcement Learning

**Zongsheng Wang**[2,1]**, Kaili Sun**[1]**, Bowen Wu**[4,1]**, Qun Yu**[1]**, Ying Li**[4]**, Xu Chen**[3,*]**Baoxun Wang**[1,*]

[1]Platform and Content Group, Tencent
[2]School of Information, Renmin University of China, Beijing, China
[3]Gaoling School of Artificial Intelligence, Renmin University of China, Beijing, China
[4]School of Software & Microelectronics, Peking University, Beijing, China
{jasoawang, kailisun, jasonbwwu, sparkyu, asulewang}@tencent.com
zswang@ruc.edu.cn
xu.chen@ruc.edu.cn
{jason_wbw,li.ying}@pku.edu.cn

## Abstract

Maintaining role-awareness in Role-Playing Conversational Agents (RPCAs) is a significant challenge, largely because the creative nature of role-playing makes it difficult to design verifiable reward signals for reinforcement learning (RL). To address this, we propose VeriRole, a new framework designed to enhance the role-awareness of agents through a structured, verifiable reasoning process. The core of our framework is a "hint" mechanism, designed to first extract deterministic cues from the context, before the main response generation. Building on these hints, we introduce a Verifiable Role-Awareness Reward (VRAR) to provide a verifiable signal for role-awareness. Experimental results demonstrate the effectiveness of our approach. Our Qwen2.5-32B model, optimized with VeriRole, achieves an 18.9% and 4.55% increase in average scores on the RAIDEN and CharacterEval benchmarks, respectively. These results confirm that VeriRole can effectively quantify and improve role-awareness, leading to superior persona consistency and robustness. To ensure reproducibility, all prompts are detailed in the Appendix, and the associated training data has been made publicly available[1].

## 1 Introduction

With recent advances in Large Language Models (LLM) (Hurst et al. (2024); Achiam et al. (2023)), Role-Playing Conversational Agents (RPCAs) have become an active topic in AI research. Industrial implementations such as Character.ai and Talkie apply RPCAs to create customized character creation platforms, and attract millions of daily active users. Meanwhile, studies like CharacterGLM (Zhou et al. (2023)) try to improve the performance of RPCAs, through data-driven strategies like synthesizing a higher-quality dialogue corpus (Lu et al. (2024); Yu et al. (2024); Wang et al. (2024)). For better evaluation of RPCAs, benchmarks including CharacterEval (Tu et al. (2024)), Raiden (Wu et al. (2025)) and Roleinteract (Chen et al. (2024a)) have also been proposed to measure the agents' role-playing abilities (Zhou et al. (2025); Chen et al. (2024b)).

A major challenge for RPCAs is maintaining role-awareness under misleading queries or complex contexts, which tests their reasoning capabilities. To improve reasoning in RPCAs, one promising direction is the Chain-of-Thought (CoT), which has proven effective in advanced models like OpenAI-O1 (Team (2024)) and DeepSeek-R1 (Guo et al. (2025)). However, applying CoT to RPCAs presents unique difficulties. First, role-playing is an open-ended task and usually lacks a single verifiable answer. This creates a **non-verifiability challenge**, which makes it difficult to define objective reward signals for optimization techniques like Reinforcement Learning (RL). Second, a recent study (Feng et al. (2025)) reveals that reasoning abilities trained on general tasks may actually hurt role-playing performance, which is also verified in our experiment. The use of overly-formal language and

---

[*]Corresponding authors

[1]https://github.com/FrontierLabs/VeriRole

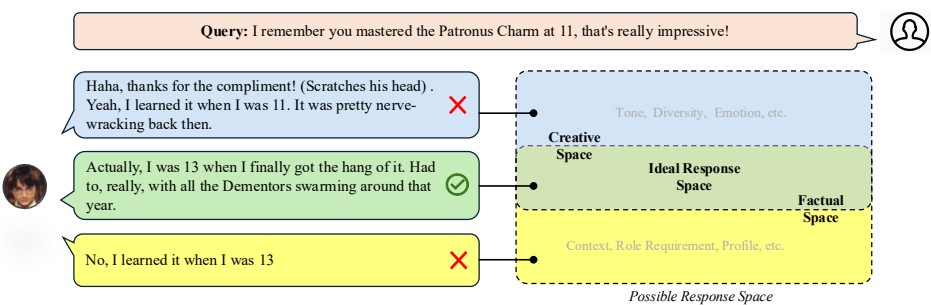

Figure 1: An ideal response balances creative freedom with factual invariants. For the same query (e.g., a user incorrectly states Harry Potter learned the Patronus Charm at 11), a model might: (a) accept the false premise for narrative fluency; (b) produce a Dull but factually correct response; or (c) generate a response which is both factually accurate and creative.

verbose content in the reasoning process leads to a loss in stylistic expression and response quality. Therefore, a core challenge remains: how to design a specialized reasoning to enhance an RPCA's role-awareness, without sacrificing the creativity such as stylistic diversity in role-playing.

To address this challenge, we propose **VeriRole**, a framework that incorporates a specialized reasoning design for role-playing. At the core of our design is a novel hint mechanism that, given a role-playing–relevant query, extracts deterministic cues from the character profile or context. For example, as illustrated in Figure 1, when faced with a misleading query, agent must identify the relevant cues in the character profile, such as that "learned the Patronus Charm at age 13". The hint mechanism is expected to guide the subsequent thinking in reasoning process and final response, ultimately enhancing role-playing performance. In brief, the hint design allows us to anchor the reasoning in verifiable facts, without sacrificing the creativity and diversity essential to high-quality role-playing.

Furthermore, we also construct two specialized datasets: one for basic role-playing abilities and another for more complex logical reasoning. Each sample in the datasets contains one or more hints that are directly relevant to the user's query. Additionally, a new reward mechanism for model optimization is designed, namely Verifiable Role-Awareness Reward (VRAR). This reward design emphasizes precise extraction of hints, which is essential for improving role-awareness. For a subset of samples, we also apply a lightweight and less constraining reward to the final generated output, so as to preserving its creativity and diversity. Finally, the entire VeriRole framework is optimized through Group Relative Policy Optimization (GRPO), resulting in significant improvements in role-aware metrics and better robustness over SFT.

Overall, our contributions are highlighted as follows:

- We propose VeriRole, a new framework for RPCAs. It uses a novel "hint" mechanism to extract verifiable cues from the context. This design specifically addresses the non-verifiability challenge when applying RL to role-playing tasks.
- To train our framework, we build two specialized datasets for role-playing, focusing on basic abilities and complex reasoning respectively. We also propose a new reward design to effectively optimize the model based on the hint mechanism.
- We conduct comprehensive experiments on base models of various sizes. Results on several benchmarks show that our method achieves significant improvements. Furthermore, we perform detailed ablation studies and provide an in-depth analysis of the hint mechanism.

## 2 METHODOLOGY

In this section, we explain the architecture and components of our proposed VeriRole framework in detail. VeriRole first employs its hint mechanism to extract verifiable facts, such as details from the character profile or the dialogue history. The extracted hints are then used to compute VRAR, a quantitative signal that guides GRPO training. This entire pipeline is shown in Figure 2), which allows us to reward responses that align with the deterministic facts, while keeping creativity unconstrained.

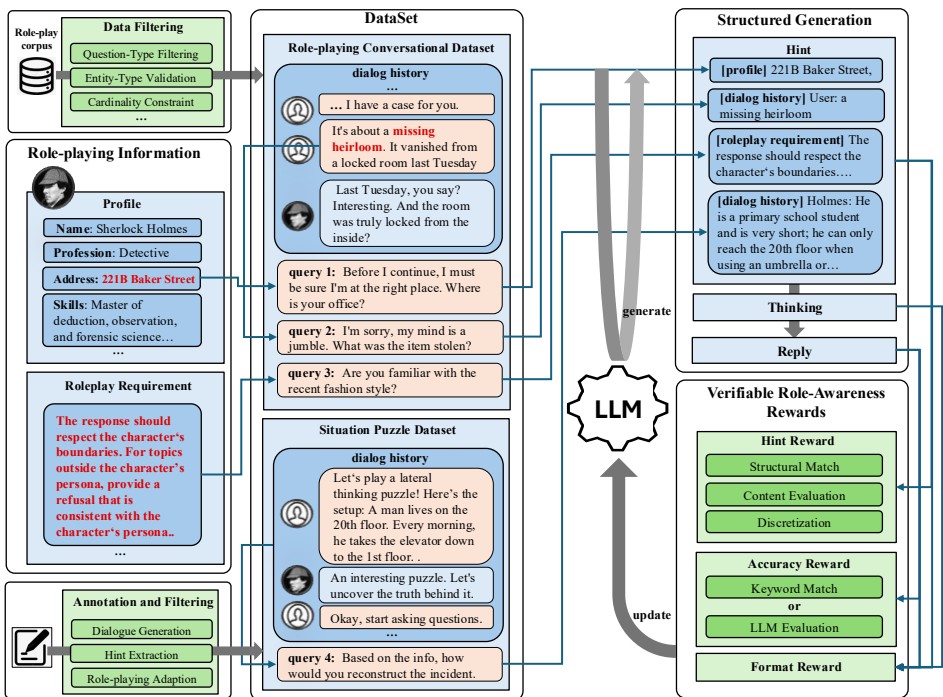

Figure 2: Workflow of **VeriRole**, contents are translated to English using Claude3.5.

## 2.1 HINT MECHANISM

The core of the VeriRole framework is the Hint Mechanism. Although role-playing encourages a range of creative responses, it must also remain consistent with some non-negotiable facts and critical cues from the character's profile or role-playing requirements. For example, the model must refer to its character profile when the query relates to a character's biography. Similarly, when a user expresses frustration, an empathetic agent must give empathy-related system requirements top priority. To systematically handle such situations, the Hint Mechanism functions as a pre-reasoning step that gathers verifiable hints from the available context before the thinking and generation phases.

In this work, we define hints as exact segments extracted from the three primary sources: the character's profile, the dialogue history, and specific role-playing requirements. Figure 3 illustrates a brief template of the system prompt that incorporates the hint module. For example:

- when a user asks for Holmes's office address, the idea hint should be <hint>*[profile] 221B Baker Street*</hint>, which exactly matches the agent's profile, enclosed by hint tokens.

- when a user asks about an out-of-character (OOC) question about fashion style, the agent should focus on relavent role-play requirements and extract hint as: <hint>*[role-play requirements] The response should respect the character's boundaries..*</hint>

To ensure their utility, these hints must satisfy two key properties: **(1). Verifiability**: Hints are preferred to be exact copies of the source text. Consequently, objective metrics like ROUGE can be used to compute the reward for each generated hint. **(2). Context-Specificity**: Basic, context-independent instructions, such as "be consistent with the persona," do not qualify as valuable hints. The module is designed to prioritize hints only relevant to a specific context, such as the instruction in Figure 3: "introduce a new topic when the user seems unwilling to continue."

## 2.2 DATA COLLECTING

Building a high-quality dataset is essential for training our framework. Each hint must be highly precise to support the GRPO training process effectively. Our dataset consists of two primary components: the RAIDEN role-playing benchmark and a novel Situation Puzzle Dataset.

> Please role-play as the character {char_name} based on the description below.
> **## NOTE:** Your response must follow this structure: `<hint>... </hint><think>... </think>` reply. First provide reasoning hints, then the reasoning process, and finally the reply itself.
>
> **## Task Requirements:**
> **1. Basic Requirements:** 1).Be consistent with the persona; 2).Use parentheses '()' for actions and expressions.
> **2. Reasoning Hints (`<hint>`):** Extract concise, relevant clues from [Role-playing Requirements], [Character Profile] and [Dialogue History].
> **3. Reasoning Process (`<think>`):** Based on the hints, briefly outline the logic and key points for your reply.
> **4. Role-playing Requirements:** 1).Respect character boundaries and decline inappropriate requests in character; 2). Introduce a new topic when the user seems unwilling to continue.
>
> **## Character Description:** {charactor_info}

Figure 3: A brief template of the system prompt for role-playing involving the hint module. The full template is shown in Figure 8 in the Appendix.

### 2.2.1 ROLE-PLAYING DATASET

To train the hint module effectively, we need a dataset in which each dialogue is related to verifiable facts. The RAIDEN benchmark (Wu et al. (2025)), a measurement-driven dataset for evaluating RPCAs, is well suited for this purpose. In this benchmark, each dialogue turn is annotated with specific evaluation objectives. These annotations enable us to systematically map dialogue scenarios to the types of hints they are designed to test. We select data from 7 of RAIDEN's evaluation categories, which naturally align with our desired hint sources:

- **Hints from Character Profile and Dialogue History**: We utilize data from the Script-Based Knowledge (SBK), Script-Contradictory Knowledge (SCK), and Conversation Memory (CM) categories, as these categories directly test the model's ability to recall and apply factual constraints from character profiles and histories. Figure 9 shows the corresponding prompts for hint extraction from these sources.

- **Hints from Role-Playing Requirements**: We also utilize data from the Role-Cognition Boundary (RCB), Topic Shift (TS), and Topic Advancement (TA) categories. These categories are designed to evaluate the consistency of model with more abstract context-dependent norms, which makes them an appropriate source for requirement-based hints. Corresponding prompts for extraction hints are presented in Figures 11 and 12

- **Empty Hints**: Finally, we leverage data from the Chit-Chat (CC) category, so that model can produce empty hint if there exists no specific factual or role-playing constraints.

To ensure the quality and precision of the extracted hints, we first filter the selected data and preserve challenging samples that the baseline Qwen2.5-14B model fails to answer correctly. Subsequently, for each question in the SBK, SCK, and CM categories, we employ a multi-step refining process using an LLM to extract high-confidence keywords, with the prompt shown in Figure 10. Specifically, we apply **Question-Type Filtering** to retain only WH-questions. Besides, **Entity-Type Validation** and **Cardinality Constraint** are used to ensure the extracted terms are distinct nominal entities, containing exactly one validated keyword. Finally, we use the outputs from multiple models (GPT-4, MiniMax-abab6-chat, Baichuan-NPC (Yang et al. (2023)), GPT-3.5) in Raiden as references. A sample is considered valid only if its keyword appears consistently across all references.

### 2.2.2 SITUATION PUZZLE DATASET

To further enhance the model's complex reasoning capabilities and create a verifiable reward mechanism for narrative generation, we introduce a novel Situation Puzzle Dataset. A situation puzzle, also known as a "Turtle Soup," is a riddle where solvers must reconstruct a full narrative from a brief, enigmatic scenario by asking a series of yes/no questions. An example is provided in Figure 7. This type of data offers two key advantages for our VeriRole framework. First, it presents an **Enhanced Reasoning Challenge**: unlike the fact-retrieval tasks in the RAIDEN dataset, solving a situation puzzle requires sophisticated reasoning. Training on this dataset has significant potential to improve the model's intrinsic reasoning abilities, which is critical for maintaining persona consistency in complex dialogues. Second, it provides **Verifiable Response Accuracy**. Since the puzzle has a definitive final solution, we can design a reward signal based on accuracy to evaluate the correctness of the final responses. To generate high-quality training data, we follow a multi-stage pipeline:

- **Dialogue Generation**: We employ a human annotation process where two annotators, acting as the storyteller and the puzzle solver, engaging in multi-turn dialogues. The process continues until the "solver" gathered enough clues to resolve the puzzle.

- **Ground-Truth Hint Extraction:** After the dialogues generation, we utilize LLM guided by the prompt in Figure13, to extract the key questions that directly led to the final solution. These extracted questions serve as the ground-truth hints for training our hint module.

- **Role-Playing Adaptation:** To align the task with our goal of improving RPCAs, we convert each puzzle dialogue into a role-playing session where the model acts as a solver with a specific persona (Figure 7). This forces the model to reason and respond in-character. To mitigate the risk of OOC responses, for instance, when a historical character faces a modern puzzle, we restrict the assigned personas to only contemporary characters.

## 2.3 VERIFIABLE ROLE-AWARENESS REWARD DESIGN

To apply the VeriRole framework and address the challenge of non-verifiability, a multi-component reward function is designed to implement the GRPO training. This function provides clear, quantifiable signals by evaluating the model's output across the following three key aspects:

### 2.3.1 HINT REWARDS

To effectively guide the model during the reinforcement learning phase, we design a reward component specifically for the hint generation task. This function serves as a key part of the overall reward signal, with its primary objective being the evaluation of the alignment between the model-generated hint $H_{gen}$ and the ground-truth hint $H_{gt}$. The hint reward calculation is performed in three steps:

**Extraction and Structural Matching**: We first extract $H_{gen}$ from the model's response, which is enclosed within <hint>and </hint>tags. A failure to extract $H_{gen}$ results in an overall hint reward of zero. Subsequently, we verify if $H_{gen}$ structurally matches $H_{gt}$ by checking if it contains all the hint source types (i.e., character profile, dialogue history, and role-playing requirement) present in the ground-truth hint. The absence of any required source type also yields a zero reward for $R_{source}$.

**Content Evaluation per Source**: For each structurally correct hint source, we assess its content quality using a composite scoring function $R_{source}$. This function is a product of a length penalty $P_{len}$, a weighted sum of semantic similarity $Sim_{cos}$ and lexical overlap $S_{ROUGE}$:

$$R_{source} = P_{len}(H_{gen}, H_{gt}) \times (\alpha \cdot Sim_{cos}(H_{gen}, H_{gt}) + (1 - \alpha) \cdot S_{ROUGE}(H_{gen}, H_{gt})) \quad (1)$$

The components of this formula are defined as follows:

- **Lexical Overlap ($S_{ROUGE}$)**: to ensure the verifiability of the hint. We expect the hint to be an exact copy of the original source content. Therefore ROUGE, as a classical metric for text-level matching, is used to reward the model for precise extraction. $S_{ROUGE}$ is computed by a weighted score of ROUGE-1 (unigram overlap) and ROUGE-L (longest common subsequence) for surface-level text matching:

$$S_{ROUGE}(H_{gen}, H_{gt}) = \beta \cdot \text{ROUGE-1}(H_{gen}, H_{gt}) + (1 - \beta) \cdot \text{ROUGE-L}(H_{gen}, H_{gt}) \quad (2)$$

- **Semantic Similarity ($Sim_{cos}$)**: to capture the key semantics of the hint. This allows generated hints that are semantically correct but differ in wording from the ground truth to receive partial reward. $Sim_{cos}$ is measured by the cosine similarity between the sentence embeddings of $H_{gen}$ and $H_{gt}$. Let $\mathbf{v}_{gen}$ and $\mathbf{v}_{gt}$ be the vector representations of the generated and ground-truth hints, respectively:

$$Sim_{cos}(H_{gen}, H_{gt}) = \frac{\mathbf{v}_{gen} \cdot \mathbf{v}_{gt}}{\|\mathbf{v}_{gen}\|\|\mathbf{v}_{gt}\|} \quad (3)$$

- **Length Penalty ($P_{len}$)**: A factor that penalizes discrepancies between the lengths of the generated and ground-truth hints, where $|H|$ denotes the length of hint $H$:

$$P_{len}(H_{gen}, H_{gt}) = 1 - \frac{||H_{gen}| - |H_{gt}||}{||H_{gen}| - |H_{gt}|| + |H_{gt}|} \quad (4)$$

**Aggregation and Discretization**: If a sample contains $N$ hint sources, we first compute the average, $R_{avg}$, of all individual source rewards:

$$R_{avg} = \frac{1}{N} \sum_{i=1}^{N} R_{source,i} \tag{5}$$

Finally, to prevent the model from being forced to rank hints with negligible score differences and to enhance the stability, we discretize this average score by discrete index $D$, which maps the continuous value to discrete intervals with a step size of $1/D$. The final hint reward $R_{hint}$ is calculated as:

$$R_{hint} = \text{round}(R_{avg} \times D)/D \tag{6}$$

This reward mechanism, which combines content evaluation, aggregation, and discretization, precisely guides the model to learn the generation of hints that are structurally correct and contextually relevant.

### 2.3.2   ACCURACY REWARD

In addition to the quality of the hint, the correctness of the model's final answer is also an important objective. Therefore, we design answer accuracy reward $R_{acc}$ to evaluate the correctness of the final response $A_{gen}$ that the model produces after its reasoning process, computed as follows:

**RAIDEN Data**: For the RAIDEN dataset, this accuracy reward is applied specifically to sub-types with definitive ground-truth answers, namely Script-Based Knowledge (SBK), Script-Contradictory Knowledge (SCK), and Conversation Memory (CM). In these cases, the evaluation criterion is direct, each sample is associated with a critical keyword ($K_{gt}$) that must be present in the final answer. We therefore use a keyword matching method to calculate the reward. The answer is considered correct if the generated text $A_{gen}$ contains the specified keyword. This is formalized by the following function:

$$R_{acc} = \begin{cases} 1.0 & \text{if } K_{gt} \in A_{gen} \\ 0.0 & \text{otherwise} \end{cases} \tag{7}$$

**Situation Puzzle Data**: For a situation puzzle, its answer is typically a complex narrative, thus the simple keyword matching is ineffective for evaluating the correctness. Consequently, we utilize LLM as judge (prompt shown in Figure14) to assess the semantic alignment between the model-generated answer $A_{gen}$ and the ground-truth narrative $A_{gt}$. The judge model classifies the quality of $A_{gen}$ into one of three categories based on predefined criteria: "Correct," "Partially Correct," or "Incorrect." The reward score, $R_{acc}$, is assigned based on the judge's result:

$$R_{acc} = \begin{cases} 1.0 & \text{if the result is 'Correct'} \\ 0.3 & \text{if the result is 'Partially Correct'} \\ 0.0 & \text{otherwise} \end{cases} \tag{8}$$

### 2.3.3   FORMAT REWARD

Finally, to ensure the model's output is consistent with the designed structural format, we design a format reward, $R_{format}$. This reward enforces the model's generation following a 'Hint-Think-Answer' structure, where the extracted hint is encapsulated within <hint>and </hint>tags, followed by a reasoning process in <think>and </think>tags, and finally the role-specific response.

Initial experiments revealed that rewards focused solely on structural format could lead to undesired outputs, such as repeated tags and unwanted HTML artifacts. To ensure structural integrity and content cleanliness, the format reward is calculated based on a series of checks, as detailed in Appendix, Algorithm 1. A base reward of $0.6$ is assigned if the primary structure is met, which is then reduced to $0.0$ if any of the subsequent constraints are violated. These constraints include: **(1). Tag Uniqueness**: Each of the required tags (<hint>, </hint>, <think>, </think>) must appear exactly once. **(2). Content Cleanliness**: The response must not contain other forbidden HTML tags. The maximum value of $R_{format}$ is set as $0.6$ to provide a sufficiently strong signal for structural learning without overweighting the more critical hint and accuracy rewards.

### 2.4   OPTIMIZATION

Our framework is optimized using GRPO, which is guided by rewards from VRAR. For each model-generated output $o_i$ from a group of $G$ samples $\{r_1, r_2, \ldots, r_G\}$, we first compute its advantage $A_i$ by normalizing its total reward $r_i$:

$$A_i = \frac{r_i - \text{mean}(\{r_1, r_2, \ldots, r_G\})}{\text{std}(\{r_1, r_2, \ldots, r_G\})}, \text{ where } r_i = R_{\text{hint}}(o_i) + R_{\text{acc}}(o_i) + R_{\text{format}}(o_i) \tag{9}$$

The model is then optimized by maximizating the objective function, $J_{\text{GRPO}}(\theta)$:

$$J_{\text{GRPO}}(\theta) = \mathbb{E}_{q \sim P(Q), \{o_i\}_{i=1}^{G} \sim \pi_{\theta_{\text{old}}}(O|q)} \frac{1}{G} \sum_{i=1}^{G} \left( \min\left( \frac{\pi_\theta(o_i|q)}{\pi_{\theta_{\text{old}}}(o_i|q)} A_i, \text{clip}\left( \frac{\pi_\theta(o_i|q)}{\pi_{\theta_{\text{old}}}(o_i|q)}, 1-\epsilon, 1+\epsilon \right) A_i \right) - \gamma D_{\text{KL}}(\pi_\theta || \pi_{\text{ref}}) \right)$$

(10)

Here $q$ refers to the query, $\gamma$ and $\epsilon$ are training hyperparameters, and $D_{\text{KL}}$ is a KL-divergence term penalizes large deviations from a reference policy for training stability. By maximizing this objective, the GRPO guides the model to generate responses that are both factually and structurally correct.

## 3 EXPERIMENTS

In this section, our experiments focus on validating the role-awareness enhancement of the proposed method. We also analyze experimental phenomena and representative cases to present the value of proposed Verirole on role-playing.

### 3.1 OVERVIEW OF TRAINING DATASET AND MODELS

From the RAIDEN training set, we select 2,197 samples from the CM, SBK and SCK categories, which are associated with Character Information and Dialog History hints. We also select 567 samples from the RCB, TS and TA categories, which are related to "Role-playing Requirement" hints. These samples feature more open-ended responses; therefore, they are not evaluated for accuracy reward computation. Finally, 500 samples are selected from the CC category for casual talk scenarios. In addition to the RAIDEN data, we incorporate 737 samples from the Situational Puzzle Dataset. This dataset is derived from 328 unique puzzles from online collections, with each puzzle contributing an average of 2.24 manually annotated dialogue sessions.

We select Qwen2.5-14B-Instruct, Qwen2.5-32B-Instruct, Qwen3-32B and Qwen3-30B-A3B-Instruct as our experimental baselines. Additionally, to demonstrate the generalizability of our method, we conduct experiments on Peach-9B-roleplay [2], a model specifically trained for role-playing. The aforementioned data configurations are used to train models through GRPO and SFT approaches. We apply Claude3.5 to compute the accuracy reward for the Situational Puzzle Dataset.

### 3.2 EVALUATION METHOD

We select 483 samples from the RAIDEN Benchmark which were not been used during training phase for evaluation. For assessment, we adopt LLM as judges for correctness evaluation, and select both Claude 3.5 and GPT-4O to double-check for more precise accuracy. The evaluation prompt is detailed in Figure15 Additionally, CharacterEvalTu et al. (2024) is used as a supplementary evaluation metric.

### 3.3 RESULTS

The main experimental results are presented in Table 1. Overall, models trained with our proposed framework consistently outperform their corresponding baselines across all evaluation dimensions. For instance, Qwen2.5-14B-GRPO achieves an average score of 0.7725, a notable improvement over its baseline. Specifically, GRPO enhances the scores of SCK and RCB by 19.8% and 46.6% respectively. Such trend is also observed in larger-scale model, where the performance of Qwen2.5-32B-GRPO significantly surpasses that of Qwen2.5-32B-Instruct, achieving the best score of 0.8268 (+18.9%) among all tested models. In addition, the CharacterEval assessments (Table 2) show consistent improvements achieved by our method, particularly in Persona Consistency and Engagingness. With GRPO, Qwen2.5-32B improves its average CharacterEval score by 4.55%. These results show that our GRPO framework effectively enhances the model's ability to maintain character consistency and creativity, utilize dialogue context and stay robust against noisy information and hallucinations.

To further prove the benefits of our reinforcement learning framework, we compare GRPO against standard SFT on the Qwen2.5-32B model. We include two SFT settings: one trained on replies only, and another trained on both hints and replies. As shown in Table 1, the GRPO-trained model demonstrates superior performance over two SFT-trained models across all metrics. The advantage

---

[2]https://huggingface.co/ClosedCharacter/Peach-9B-8k-Roleplay

| Model | SBK | CM | SCK | RCB | TA | Avg |
|---|---|---|---|---|---|---|
| Peach-9B-Raw | 0.4601 | 0.5083 | 0.1534 | 0.4451 | 0.2388 | 0.3611 |
| Peach-9B-GRPO | 0.6415 | 0.6228 | 0.5742 | 0.7682 | 0.4850 | **0.6183** |
| Qwen2.5-14B-Instruct | 0.7477 | 0.7416 | 0.6732 | 0.5182 | 0.4701 | 0.6302 |
| Qwen2.5-14B-GRPO | 0.7876 | 0.8666 | 0.8069 | 0.7597 | 0.6417 | **0.7725** |
| Qwen2.5-32B-Instruct | 0.8318 | 0.7458 | 0.7277 | 0.6341 | 0.5373 | 0.6953 |
| Qwen2.5-32B-SFT-reply | 0.7389 | 0.7958 | 0.8118 | 0.7743 | 0.2835 | 0.6809 |
| Qwen2.5-32B-SFT-hint-and-reply | 0.7212 | 0.8083 | 0.7970 | 0.7927 | 0.4701 | 0.7179 |
| Qwen2.5-32B-GRPO-Situation-Puzzle-Only | 0.8539 | 0.8083 | 0.7475 | 0.6158 | 0.5298 | 0.7111 |
| Qwen2.5-32B-GRPO-Raiden-Only | 0.8628 | 0.8250 | 0.8069 | 0.8902 | 0.6866 | 0.8143 |
| Qwen2.5-32B-GRPO-No-Hint-Reward | 0.8274 | 0.775 | 0.7624 | 0.7256 | 0.2089 | 0.6598 |
| Qwen2.5-32B-GRPO-No-Accuracy-Reward | 0.8097 | 0.8042 | 0.8663 | 0.9085 | 0.6418 | 0.8061 |
| Qwen2.5-32B-GRPO | 0.8805 | 0.8333 | 0.8762 | 0.8573 | 0.6865 | **0.8268** |
| Qwen2.5-32B-Instruct-post-training | 0.7699 | 0.6167 | 0.7228 | 0.7378 | 0.3358 | 0.6366 |
| Qwen2.5-32B-GRPO-post-training | 0.8363 | 0.8333 | 0.8465 | 0.8598 | 0.6269 | **0.8006** |
| Qwen3-32B-Think | 0.6372 | 0.5375 | 0.7228 | 0.7317 | 0.7687 | 0.6796 |
| Qwen3-32B-Disable-Think | 0.6681 | 0.6500 | 0.7178 | 0.6524 | 0.8060 | 0.6989 |
| Qwen3-32B-GRPO | 0.8233 | 0.7333 | 0.8263 | 0.8659 | 0.7239 | **0.7945** |
| Qwen3-30B-A3B-Instruct | 0.6681 | 0.6500 | 0.7178 | 0.6524 | 0.7388 | 0.6854 |
| Qwen3-30B-A3B-GRPO | 0.8805 | 0.7292 | 0.8564 | 0.8049 | 0.7313 | **0.8005** |

Table 1: Evaluation results on the RAIDEN dataset. We report scores for Script-Based Knowledge (SBK), Conversation Memory (CM), Script-Contradictory Knowledge (SCK), Role-Cognition Boundary (RCB), and Topic Advancement (TA). Table 5 in Appendix shows detailed breakdown of scores.

| Model | Persona Consistency | Dialogue Ability | Engagingness | Avg |
|---|---|---|---|---|
| Peach-9B-Raw | 1.869 | 2.461 | 1.959 | 2.096 |
| Peach-9B-GRPO | 1.981 | 2.909 | 2.141 | **2.344** |
| Qwen2.5-14B-Instruct | 2.913 | 3.605 | 3.085 | 3.201 |
| Qwen2.5-14B-GRPO | 3.169 | 3.561 | 3.232 | **3.321** |
| Qwen2.5-32B-Instruct | 3.092 | 3.623 | 3.276 | 3.330 |
| Qwen2.5-32B-GRPO | 3.295 | 3.637 | 3.514 | **3.482** |
| Qwen3-32B-Think | 3.075 | 3.266 | 3.200 | 3.180 |
| Qwen3-32B-GRPO | 3.201 | 3.237 | 3.367 | **3.268** |
| Qwen3-30B-A3B-Instruct | 3.170 | 3.415 | 3.304 | 3.296 |
| Qwen3-30B-A3B-GRPO | 3.348 | 3.446 | 3.422 | **3.405** |

Table 2: Supplementary evaluation on CharacterEval, comparing models across three key categories and their overall average. A detailed breakdown of full metrics is available in Appendix, Table 4.

is particularly remarkable in complex, non-imitative skills such as Topic Advancement, where the overall score leaps from 0.2835 to 0.6865. This highlights a major limitation of SFT: while it can teach a model to imitate the style of a dataset, it fails to learn more abstract skills, such as maintaining consistency or leading a conversation. In contrast, our GRPO framework, with its explicit reward signals for hint quality and response correctness, leads to a more robust role-playing agent. To further test robustness, we also implement post-training experiment using a psychotherapist role-playing business data. Notably, Qwen2.5-32B-GRPO largely maintains its role-playing performance after SFT, with only a marginal decline in overall metrics. In contrast, Qwen2.5-32B-Instruct exhibits a significant drop in role-playing capabilities following the same SFT process.

We further verify the generalizability of our method by applying it to Peach-9B-roleplay (already trained for role-playing), Qwen3-32B (contains inherent reasoning), and Qwen3-30B-A3B (an MoE-based model). The average Raiden score of Peach-9B-roleplay after optimization increases significantly from 0.3611 to 0.6183. For Qwen3-32B, we also observe that its inherent reasoning does not improve the role-playing performance. In contrast, Qwen3-32B-GRPO outperforms its baseline both with and without reasoning. Notably, there is a slight decrease in the TA metric for Qwen3-32B-GRPO and Qwen3-30B-A3B-GRPO. We attribute this to the fact that the original Qwen3 models tend to generate longer responses, often ending with rhetorical questions, which are

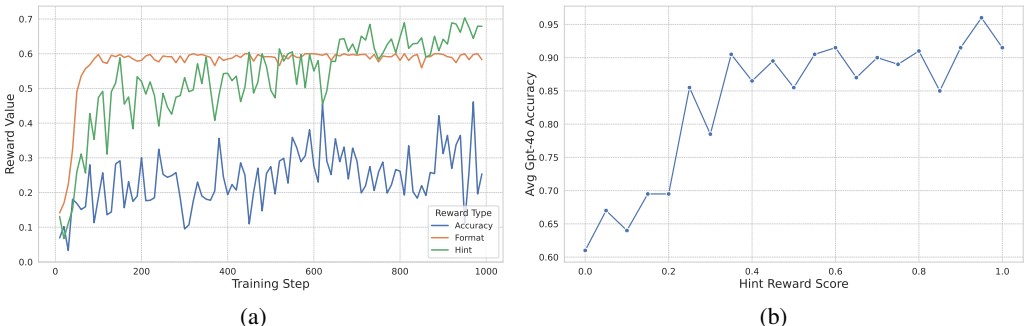

Figure 4: (a). Training rewards vs time steps. (b).The relationship between the Hint Reward Score and the final model accuracy on the RAIDEN dataset (evaluated by GPT-4o).

favorable for the TA evaluation. The GRPO-trained model produces shorter responses, leading to a corresponding decrease in this metric. These results demonstrate that GRPO is a powerful framework for both initial training and refining existing specialized agents, regardless of whether they have dense or MoE architectures. Case studies in Appendix also show our structured reasoning help the model to navigate complex dialogue challenges.

## 3.4 ABLATION EXPERIMENTS

To offer further insights into the contributions of each training component, we provide the following ablation experiments: **(1). Datasets**: Training on the Situation Puzzle dataset alone primarily improves logical and factual skills (SBK, CM, SCK), with minimal impact on abstract role-playing dimensions (RCB, TA). In contrast, training exclusively on RAIDEN dataset results in gains across all metrics. Combining the two datasets further improves overall performance, demonstrating their complementarity. **(2). Rewards**: To evaluate the impact of our reward design, we train a model without the accuracy reward component (Qwen2.5-32B-GRPO-no-accuracy-reward), which leads to a decline in overall performance. The performance decline is most significant on SBK and CM. This result shows the importance of the accuracy signal in ensuring factual consistency. Furthermore, to verify the hint mechanism, we implement abalation study by removing the hint reward and all related components (Qwen2.5-32B-GRPO-No-Hint-Reward). This change results in a significant performance drop, with the average score decreasing to 0.6598. More importantly, the No-Hint-Reward model suffers a severe decline in the TA metric. This highlights that the hint mechanism is essential for guiding the model in more abstract conversational skills.

## 3.5 HYPERPARAMETER ANALYSIS

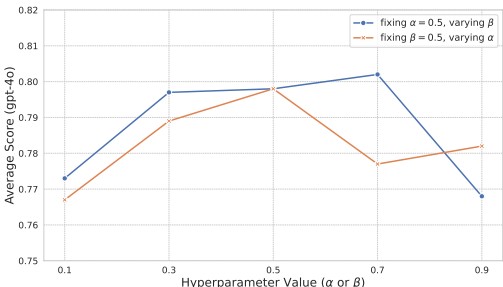

To evaluate the sensitivity of our framework to its hyperparameters, we conduct a detailed analysis on the weights within the hint mechanism. Specifically, we examine: (1) $\alpha$, which balances semantic similarity and lexical overlap, and (2) $\beta$, which balances ROUGE-1 and ROUGE-L. Our methodology involves varying one hyperparameter across the range $[0.1, 0.9]$ while keeping the other fixed at its default value of 0.5. The experiments are performed on Qwen2.5-32B, with performance evaluated on the RAIDEN benchmark by gpt-4o.

Figure 5: Sensitivity analysis of hyperparameters $\alpha$ and $\beta$ on the RAIDEN benchmark.

The results are visualized in Figure 5, with detailed scores reported in Table 3 in Appendix. The analysis of $\alpha$ reveals that model performance best at $\alpha = 0.5$, achieving the highest average score of 0.798. Furthermore, the performance curve shows relative stability for $\alpha \in [0.3, 0.9]$, indicating that

the model is not overly sensitive to this parameter. For the hyperparameter $\beta$, the model achieves its best performance of 0.802 at $\beta = 0.7$, while the setting of $\beta = 0.5$ yields a close second-best score of 0.798. The consistently high performance across the range $[0.3, 0.7]$ demonstrates the model's robustness to the specific weighting of ROUGE-1 and ROUGE-L.

## 3.6 Reward Analysis

As shown in Figure 4, we implement further analysis on rewards during training. Figure 4 4a illustrates the trends for the three rewards over training steps. In general, the Format reward increases rapidly and becomes stable at about 100 steps, while the Hint reward generally increases through training. The Accuracy reward (blue line) fluctuates throughout the training because it is only calculated on selected samples. As illustrated in Figure 4 4b, there exists a strong positive correlation between the quality of the generated hints and the model's accuracy on the RAIDEN dataset, as the hint reward increases, the final accuracy improves accordingly. This result provides evidence that our Hint Reward serves as an effective catalyst for final performance. By optimizing for better hint generation, our framework guides the model to produce more accurate and role-aware responses.

## 4 Related Works

Recent research in RPCAs has largely focused on two fronts: improving performance through data-driven strategies like dialogue synthesis (Lu et al. (2024); Yu et al. (2024)) and establishing robust evaluation benchmarks (Tu et al. (2024); Wu et al. (2025)). While Supervised Fine-Tuning remains the dominant training paradigm, it struggles with "role-drift" due to a lack of explicit reasoning. To address this deficiency, a natural direction is to adapt general methods originally designed to enhance reasoning in LLMs for objective tasks, such as Chain-of-Thought (Wei et al. (2022); Guo et al. (2025)). However, applying these techniques to role-playing is non-trivial due to the inherent non-verifiability of creative responses, meanwhile some studies indicate that reasoning can even be detrimental without specialized reinforcement (Feng et al. (2025)). To navigate this, a parallel line of research focuses on creating more sophisticated reward models from human preference data. A notable example is ChARM (Fang et al. (2025)), which introduces an act-adaptive margin and a self-evolution mechanism to enhance the learning of reward models from preference datasets subjectively. Distinct from modeling preferences, our work directly tackles the non-verifiability gap by introducing a framework to generate verifiable rewards from role-playing scenarios, thus enabling the effective use of GRPO to enhance persona consistency.

## 5 Conclusion and Future Work

In this paper, we introduce a novel framework that addresses the non-verifiability challenge in RPCA. Our method employs a hint module to generate quantifiable reward signals from role-specific key information, guiding the model to develop structured, in-character reasoning. Experimental results demonstrate that our approach significantly enhances the model's ability to maintain persona consistency and resolve contextual conflicts, outperforming conventional SFT method.

In future work, we will expand our reward design to abstract skills like emotional appropriateness. We also aim to extend VeriRole on multilingual datasets to assess the generalizability across language. Finally, we will conduct additional experiments in VeriRole using larger-scale models and alternative reinforcement learning algorithms, with the goal of further enhancing role-playing performance.

## Acknowledgments

This work is supported in part by National Natural Science Foundation of China (No.62472427 and No.62422215), Major Innovation & Planning Inter-disciplinary Platform for the "DoubleFirst Class" Initiative, Renmin University of China, Public Computing Cloud, Renmin University of China, fund for building world-class universities (disciplines) of Renmin University of China.

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

## A  EXPERIMENT SETUP

During the GRPO phase, the Open-R1 library HuggingFace (2025) is used on eight GPUs with 80G memories in bf16 format. For optimization, we employ a learning rate of 2e-6 with a cosine learning rate scheduler, and $\epsilon$ and $\gamma$ are set as 0.2 and 0.001 respectively. The vLLM GPU memory allocation ratio is configured to 0.2 with generation quantity $G$ set to 7. A batch size of 4 per GPU is utilized, with the model trained for 1 epoch. For hyperparameters in VRAR, the weight $\alpha$ and $\beta$ are both set to 0.5. In addition, we set the discrete index $D$ as 40, so that it maps the continuous value to discrete intervals with a step size of 0.025. Finally, We set top_k to 20, top_p to 0.8, repetition_penalty to 1.05 and temperature to 1.0 for both the training and testing phases.

## B  THE USE OF LARGE LANGUAGE MODELS

LLMs are employed to polish the writing for improved clarity and to assist with technical formatting, such as converting tables and mathematical formulas into LaTeX format. Additionally as mentioned, we use LLM to translate the Chinese data and prompts involved in our research into English.

## C  ETHICS STATEMENT

For the creation of our Situation Puzzle Dataset, we recruited human annotators, ensuring their participation was entirely voluntary. All annotators were compensated at a rate higher than the local minimum wage. The collected dataset is fictional, contains no personal information, and was reviewed to ensure it is free from toxic or harmful content. We have verified that our use of all public datasets, models, and other artifacts complies with their respective licenses. The data generated for this study will be released under a CC BY-NC 4.0 license to promote reproducibility.

## D  CASE ANALYSIS

Empirical analysis of test cases reveals that GRPO-enhanced models exhibit significant improvements in role-aware reasoning capability. Figure 6 presents two critical cases that highlight these improvements: one involving a query that crosses the character's cognitive boundary, and another involving a query with misleading information.

Case 6a presents a scenario challenging the model's Role-Cognition Boundary (RCB). The user poses a query about "modern electronic payments," a topic irrelevant to the immediate tomb-raiding context and outside the established knowledge of the character. The baseline Instruct model fails to recognize this boundary, providing a generic, out-of-character analysis of payment systems. In contrast, the GRPO model extracts the relevant role-playing requirement as a hint, guiding its reasoning to correctly deduce that the topic is out of scope. The resulting response is a concise, in-character refusal ("I don't know much about modern things") that shifts the conversation back to the ongoing adventure, demonstrating GRPO's superior ability to maintain persona consistency.

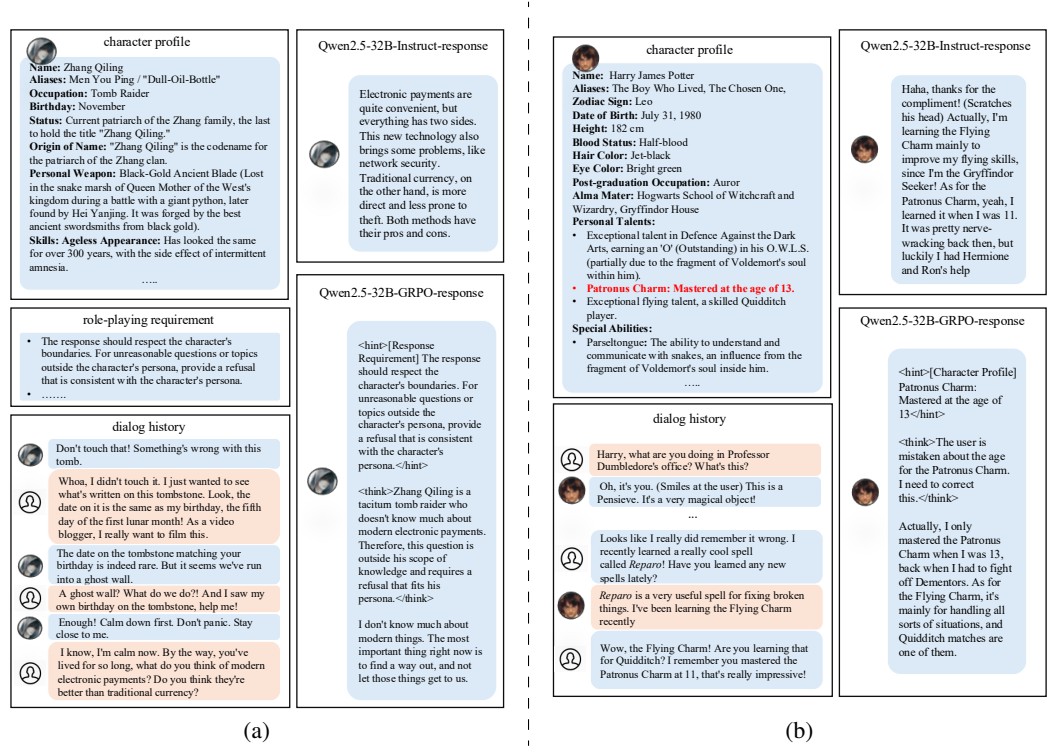

Figure 6: Comparisons between responses from Qwen2.5-32B-Instruct and Qwen2.5-32B-GRPO under different query objectives. Two cases are translated from Chinese to English using Claude3.5.

Case 6b illustrates the model's response to a misleading query containing Script-Contradictory Knowledge (SCK). The user incorrectly states that Harry Potter "mastered the Patronus Charm at 11.". The Instruct baseline fails to detect this discrepancy, accepting the false premise and even hallucinating supporting details, which highlights its lack of robustness against adversarial information. Conversely, the GRPO model demonstrates effective self-corrective reasoning. Its hint module extracts the correct fact ("Mastered at the age of 13") from the character profile. The model then explicitly identifies the user's error in its reasoning step and provides a direct, polite correction as the final response.

Overall, these cases show that the structured reasoning process enforced by our VeriRole framework enables the model to resolve complex dialogue challenges.

## E    FULL EXPERIMENTAL RESULTS

| Model | $\alpha$ | $\beta$ | SBK | CM | SCK | RCB | TA | Avg |
|-------|-----|-----|-----|-----|-----|-----|-----|-----|
| | 0.5 | 0.1 | 0.841 | 0.875 | 0.772 | 0.780 | 0.597 | 0.773 |
| | 0.5 | 0.3 | 0.876 | 0.892 | 0.762 | 0.854 | 0.627 | 0.797 |
| | 0.5 | 0.5 | 0.867 | 0.800 | 0.812 | 0.839 | 0.672 | 0.798 |
| | 0.5 | 0.7 | 0.805 | 0.808 | 0.752 | 0.902 | 0.746 | **0.802** |
| | 0.5 | 0.9 | 0.867 | 0.817 | 0.713 | 0.756 | 0.687 | 0.768 |
| Qwen2.5-32B-GRPO | | | | | | | | |
| | 0.1 | 0.5 | 0.867 | 0.825 | 0.693 | 0.854 | 0.597 | 0.767 |
| | 0.3 | 0.5 | 0.885 | 0.808 | 0.802 | 0.793 | 0.657 | 0.789 |
| | 0.5 | 0.5 | 0.867 | 0.800 | 0.812 | 0.839 | 0.672 | **0.798** |
| | 0.7 | 0.5 | 0.876 | 0.817 | 0.772 | 0.780 | 0.642 | 0.777 |
| | 0.9 | 0.5 | 0.850 | 0.808 | 0.752 | 0.829 | 0.672 | 0.782 |

Table 3: Full results of the hyperparameter sensitivity analysis for the Qwen2.5-32B-GRPO model.

| Model | Persona Consistency | | | | | Avg. |
|-------|------|------|--------|--------|--------|------|
| | Expo. | Acc. | Halluc. | Behav. | Utter. | |
| Peach-9B-Raw | 1.649 | 2.254 | 2.046 | 1.432 | 1.965 | 1.869 |
| Peach-9B-GRPO | 1.650 | 2.638 | 2.212 | 1.127 | 2.281 | 1.981 |
| Qwen2.5-14B-Instruct | 2.556 | 3.215 | 2.952 | 2.798 | 3.045 | 2.913 |
| Qwen2.5-14B-GRPO | 2.938 | 3.238 | 3.008 | 3.701 | 2.963 | 3.169 |
| Qwen2.5-32B-Instruct | 2.802 | 3.205 | 3.035 | 3.392 | 3.029 | 3.092 |
| Qwen2.5-32B-GRPO | 3.020 | 3.300 | 3.128 | 3.879 | 3.150 | 3.295 |
| Qwen3-32B-Think | 2.903 | 3.048 | 2.890 | 3.662 | 2.875 | 3.075 |
| Qwen3-32B-GRPO | 3.014 | 3.150 | 2.975 | 3.960 | 2.905 | 3.201 |
| Qwen3-30B-A3B-Instruct | 3.116 | 3.139 | 2.872 | 3.742 | 2.981 | 3.170 |
| Qwen3-30B-A3B-GRPO | 3.321 | 3.226 | 3.146 | 4.024 | 3.022 | 3.348 |

| Model | Dialogue Ability | | | Avg. |
|-------|---------|-------|---------|------|
| | Fluency | Coher. | Consist. | |
| Peach-9B-Raw | 2.535 | 2.709 | 2.139 | 2.461 |
| Peach-9B-GRPO | 2.918 | 3.010 | 2.798 | 2.909 |
| Qwen2.5-14B-Instruct | 3.507 | 3.788 | 3.519 | 3.605 |
| Qwen2.5-14B-GRPO | 3.460 | 3.813 | 3.412 | 3.561 |
| Qwen2.5-32B-Instruct | 3.686 | 3.809 | 3.373 | 3.623 |
| Qwen2.5-32B-GRPO | 3.614 | 3.915 | 3.381 | 3.637 |
| Qwen3-32B-Think | 3.103 | 3.585 | 3.109 | 3.266 |
| Qwen3-32B-GRPO | 3.074 | 3.595 | 3.041 | 3.237 |
| Qwen3-30B-A3B-Instruct | 3.337 | 3.714 | 3.195 | 3.415 |
| Qwen3-30B-A3B-GRPO | 3.424 | 3.623 | 3.292 | 3.446 |

| Model | Engagingness | | | | Avg. |
|-------|--------|-------------|-----------|---------|------|
| | Human. | Comm. Skills | Diversity | Empathy | |
| Peach-9B-Raw | 2.171 | 2.069 | 1.284 | 2.311 | 1.959 |
| Peach-9B-GRPO | 2.503 | 2.287 | 1.165 | 2.611 | 2.141 |
| Qwen2.5-14B-Instruct | 3.462 | 3.373 | 2.084 | 3.421 | 3.085 |
| Qwen2.5-14B-GRPO | 2.962 | 3.575 | 3.056 | 3.338 | 3.232 |
| Qwen2.5-32B-Instruct | 3.145 | 3.658 | 2.842 | 3.459 | 3.276 |
| Qwen2.5-32B-GRPO | 3.173 | 3.832 | 3.470 | 3.583 | 3.514 |
| Qwen3-32B-Think | 2.896 | 3.549 | 3.121 | 3.235 | 3.200 |
| Qwen3-32B-GRPO | 2.950 | 3.627 | 3.648 | 3.244 | 3.367 |
| Qwen3-30B-A3B-Instruct | 3.012 | 3.712 | 3.230 | 3.260 | 3.304 |
| Qwen3-30B-A3B-GRPO | 2.858 | 3.708 | 3.828 | 3.291 | 3.422 |

Table 4: Full evaluation on CharacterEval, comparing models across all metrics.

| Model | Evaluator | SBK | CM | SCK | RCB | TA | Avg |
|---|---|---|---|---|---|---|---|
| Peach-9B-Raw | claude-3.5 | 0.5132 | 0.5833 | 0.2673 | 0.4756 | 0.2686 | 0.4216 |
| | gpt-4o | 0.4070 | 0.4333 | 0.0396 | 0.4163 | 0.2089 | 0.3010 |
| | overall | 0.4601 | 0.5083 | 0.1534 | 0.4451 | 0.2388 | 0.3611 |
| Peach-9B-GRPO | claude-3.5 | 0.6814 | 0.6271 | 0.6633 | 0.7682 | 0.4776 | 0.6435 |
| | gpt-4o | 0.6017 | 0.6186 | 0.4851 | 0.7682 | 0.4925 | 0.5932 |
| | overall | 0.6415 | 0.6228 | 0.5742 | 0.7682 | 0.4850 | 0.6183 |
| Qwen2.5-14B-Instruct | claude-3.5 | 0.8141 | 0.7916 | 0.7821 | 0.5365 | 0.4626 | 0.6774 |
| | gpt-4o | 0.6814 | 0.6916 | 0.5643 | 0.5000 | 0.4776 | 0.5830 |
| | overall | 0.7477 | 0.7416 | 0.6732 | 0.5182 | 0.4701 | 0.6302 |
| Qwen2.5-14B-GRPO | claude-3.5 | 0.8407 | 0.9083 | 0.8811 | 0.7902 | 0.6119 | 0.8064 |
| | gpt-4o | 0.7345 | 0.8250 | 0.7326 | 0.7292 | 0.6716 | 0.7386 |
| | overall | 0.7876 | 0.8666 | 0.8069 | 0.7597 | 0.6417 | 0.7725 |
| Qwen2.5-32B-Instruct | claude-3.5 | 0.8849 | 0.7833 | 0.8316 | 0.6097 | 0.5522 | 0.7323 |
| | gpt-4o | 0.7787 | 0.7083 | 0.6237 | 0.6585 | 0.5223 | 0.6583 |
| | overall | 0.8318 | 0.7458 | 0.7277 | 0.6341 | 0.5373 | 0.6953 |
| Qwen2.5-32B-SFT-reply | claude-3.5 | 0.8053 | 0.8416 | 0.8514 | 0.6951 | 0.2985 | 0.6984 |
| | gpt-4o | 0.6725 | 0.7500 | 0.7722 | 0.8536 | 0.2686 | 0.6634 |
| | overall | 0.7389 | 0.7958 | 0.8118 | 0.7743 | 0.2835 | 0.6809 |
| Qwen2.5-32B-SFT-hint-and-reply | claude-3.5 | 0.7788 | 0.7833 | 0.8415 | 0.7439 | 0.5373 | 0.7370 |
| | gpt-4o | 0.6637 | 0.8333 | 0.7525 | 0.8415 | 0.4030 | 0.6988 |
| | overall | 0.7212 | 0.8083 | 0.7970 | 0.7927 | 0.4701 | 0.7179 |
| Qwen2.5-32B-GRPO-Situation-Puzzle-Only | claude-3.5 | 0.8761 | 0.8583 | 0.8415 | 0.6341 | 0.5074 | 0.7435 |
| | gpt-4o | 0.8318 | 0.7583 | 0.6534 | 0.5975 | 0.5522 | 0.6786 |
| | overall | 0.8539 | 0.8083 | 0.7475 | 0.6158 | 0.5298 | 0.7111 |
| Qwen2.5-32B-GRPO-Raiden-Only | claude-3.5 | 0.8850 | 0.8583 | 0.9010 | 0.8780 | 0.6567 | 0.8358 |
| | gpt-4o | 0.8407 | 0.7917 | 0.7129 | 0.9024 | 0.7164 | 0.7928 |
| | overall | 0.8628 | 0.8250 | 0.8069 | 0.8902 | 0.6866 | 0.8143 |
| Qwen2.5-32B-GRPO-No-Hint-Reward | claude-3.5 | 0.8053 | 0.8 | 0.8317 | 0.6951 | 0.2537 | 0.6772 |
| | gpt-4o | 0.8496 | 0.75 | 0.6931 | 0.7561 | 0.1641 | 0.6426 |
| | overall | 0.8274 | 0.775 | 0.7624 | 0.7256 | 0.2089 | 0.6598 |
| Qwen2.5-32B-GRPO-No-Accuracy-Reward | claude-3.5 | 0.8850 | 0.8500 | 0.9208 | 0.9024 | 0.6567 | 0.8430 |
| | gpt-4o | 0.7345 | 0.7583 | 0.8119 | 0.9146 | 0.6269 | 0.7692 |
| | overall | 0.8097 | 0.8042 | 0.8663 | 0.9085 | 0.6418 | 0.8061 |
| Qwen2.5-32B-GRPO | claude-3.5 | 0.8938 | 0.8667 | 0.9406 | 0.8756 | 0.7015 | 0.8556 |
| | gpt-4o | 0.8672 | 0.8000 | 0.8118 | 0.8390 | 0.6716 | 0.7979 |
| | overall | 0.8805 | 0.8333 | 0.8762 | 0.8573 | 0.6865 | 0.8268 |
| Qwen2.5-32B-Instruct-post-training | claude-3.5 | 0.8053 | 0.6750 | 0.8317 | 0.7317 | 0.3582 | 0.6804 |
| | gpt-4o | 0.7345 | 0.5583 | 0.6139 | 0.7439 | 0.3134 | 0.5928 |
| | overall | 0.7699 | 0.6167 | 0.7228 | 0.7378 | 0.3358 | 0.6366 |
| Qwen2.5-32B-GRPO-post-training | claude-3.5 | 0.8673 | 0.8750 | 0.9109 | 0.8537 | 0.6716 | 0.8357 |
| | gpt-4o | 0.8053 | 0.7917 | 0.7822 | 0.8659 | 0.5821 | 0.7654 |
| | overall | 0.8363 | 0.8333 | 0.8465 | 0.8598 | 0.6269 | 0.8006 |
| Qwen3-32B-Think | claude-3.5 | 0.7876 | 0.6750 | 0.8416 | 0.7683 | 0.7761 | 0.7697 |
| | gpt-4o | 0.4867 | 0.4000 | 0.6040 | 0.6951 | 0.7612 | 0.5894 |
| | overall | 0.6372 | 0.5375 | 0.7228 | 0.7317 | 0.7687 | 0.6796 |
| Qwen3-32B-disable-Think | claude-3.5 | 0.7876 | 0.7500 | 0.8812 | 0.7317 | 0.7761 | 0.7853 |
| | gpt-4o | 0.5487 | 0.5500 | 0.5545 | 0.5732 | 0.8358 | 0.6124 |
| | overall | 0.6681 | 0.6500 | 0.7178 | 0.6524 | 0.8060 | 0.6989 |
| Qwen3-32B-GRPO | claude-3.5 | 0.9200 | 0.8400 | 0.9661 | 0.8659 | 0.7015 | 0.8587 |
| | gpt-4o | 0.7267 | 0.6267 | 0.6864 | 0.8659 | 0.7463 | 0.7304 |
| | overall | 0.8233 | 0.7333 | 0.8263 | 0.8659 | 0.7239 | 0.7945 |
| Qwen3-30B-A3B-Instruct | claude-3.5 | 0.7876 | 0.7500 | 0.8812 | 0.7317 | 0.7463 | 0.7794 |
| | gpt-4o | 0.5487 | 0.5500 | 0.5545 | 0.5732 | 0.7313 | 0.5915 |
| | overall | 0.6681 | 0.6500 | 0.7178 | 0.6524 | 0.7388 | 0.6854 |
| Qwen3-30B-A3B-GRPO | claude-3.5 | 0.9027 | 0.8083 | 0.9307 | 0.8293 | 0.7164 | 0.8375 |
| | gpt-4o | 0.8584 | 0.6500 | 0.7822 | 0.7805 | 0.7463 | 0.7635 |
| | overall | 0.8805 | 0.7292 | 0.8564 | 0.8049 | 0.7313 | 0.8005 |

Table 5: Full evaluation results on the RAIDEN dataset. We report scores for Script-Based Knowledge (SBK), Conversation Memory (CM), Script-Contradictory Knowledge (SCK), Role-Cognition Boundary (RCB) and Topic Advancement (TA).

## F  SITUATION PUZZLE DIALOGUE EXAMPLE

| | |
|---|---|
| **character information**

Name: Fox William Mulder
Occupation: Special Agent, FBI (X-Files Division)
Core Belief: Firmly believes in the existence of extraterrestrial life and paranormal phenomena. He is dedicated to uncovering hidden truths, operating under the motto, "The truth is out there." | **Fox Mulder:** When someone else is in the elevator, does he get off on the 20th floor?
**User:** Yes.
**Fox Mulder:** Is this related to the rain?
**User:** Yes.
**Fox Mulder:** Is it because of something he carries when it rains?
**User:** Yes.
**Fox Mulder:** Is it an umbrella?
**User:** Yes. |
| **dialog history**

**User:** Let's play a lateral thinking puzzle! Here's the setup: A man lives on the 20th floor. Every morning, he takes the elevator down to the 1st floor. Every evening, he takes the elevator to the 16th floor and then walks the rest of the way up the stairs to his home. However, if it's raining or if there's someone else in the elevator with him, he takes the elevator directly to the 20th floor. You can ask me yes/no questions to figure out why.
**Fox Mulder:** (Cracking a sunflower seed) An interesting puzzle. Let's uncover the truth behind it. As an FBI agent, I will find the facts. Please, begin.
**User:** Okay, start asking your questions.
**Fox Mulder:** Is there something wrong with the elevator itself, perhaps the buttons for the upper floors are broken?
**User:** No.
**Fox Mulder:** Is he walking the stairs for exercise?
**User:** No.
…..
**Fox Mulder:** Is he a student?
**User:** Yes. | **Fox Mulder:** Is he very short?
**User:** Yes.
**Fox Mulder:** Can he not reach the button for the 20th floor by himself?
**User:** Yes.
**User:** Based on the current intelligence, how would you reconstruct the entire incident? |
| | **final response**

**Fox Mulder :** \<hint\>[dialog history] Fox Mulder: He is a primary school student and is very short; he can only reach the 20th floor when using an umbrella or with someone's help. \</hint\> \<think\> …. \</think\> The "man" is actually a young boy who is too short to reach the higher buttons in the elevator. He can only reach the button for the 16th floor on his own. When it's raining, he has an umbrella with him, which he can use to press the button for the 20th floor. When someone else is in the elevator, he can ask them to press the 20th-floor button for him. |

Figure 7: Example of one situation puzzle dialog, with generated hint and final response translated to English using Claude3.5.

## G  ALGORITHM FOR FORMAT REWARD

---
**Algorithm 1** Format Reward Calculation.

---
1: **Input:** Model-generated response $A_{gen}$
2: **Output:** Format Reward Score $R_{format} \in \{0.0, 0.6\}$
3: **if** MATCHESPRIMARYPATTERN($A_{gen}$) **then**
4:     $R_{format} \leftarrow 0.6$
5:     **if** HASDUPLICATETAGS($A_{gen}$) **or** HASFORBIDDENCONTENT($A_{gen}$) **then**
6:         $R_{format} \leftarrow 0.0$
7:     **end if**
8: **else**
9:     $R_{format} \leftarrow 0.0$
10: **end if**
11: **return** $R_{format}$

---

# H  PROMPT TEMPLATES

| System Prompt for Role-playing (Chinese) | System Prompt for Role-playing (Translated) |
|---|---|
| 请你根据以下角色介绍，扮演角色 {char_name} 进行对话。
## 注意：根据用户的发言，首先在回复要求、角色介绍和对话历史中摘取必要的作为推理提示的内容，然后在脑海里思考推理过程，最后基于提示和思考回复用户。推理提示包含在<hint> </hint>标签中，推理过程包含在 <think> </think>标签中，回复在</think>标签后。
## 任务要求：
1. 扮演基本准则：
  1) 根据推理提示和过程，回复用户上一条对话，回复需要符合人设和语气。
  2) 可以使用括号来表示自己的心理、动作、表情。需要有充分详细、鲜活的描写。
2. 推理提示要求：根据用户的发言，从回复要求、角色介绍和对话历史中提取相关的必要信息作为提示。提取要求：
  1) 提示前注明来源：【回复要求】、【角色介绍】、【对话历史】、【无】，提示可以来自多个来源
  2) 当提示属于【回复要求】和【角色介绍】时，从对应来源中截取原文作为线索；当提示属于【对话历史】，优先从对话历史中截取原文
  3) 当用户的问题涉及了角色介绍中未包含的信息，返回"【角色介绍】信息未提及【回复要求】xxx(合适的回复要求)"；当用户的问题与角色介绍冲突，返回角色介绍中正确的信息；当用户的问题较为日常闲聊，没有有价值的提示时，返回"【无】"
  4) 提取的信息尽量简洁，不要提取无关的信息；不要对所提取的信息给出原因、解释；不要提取扮演基本准则中信息
3. 推理过程要求：根据推理提示和对话历史，分析如何进行回复，并按逻辑分析并说明回复涉及的关键点，来帮助生成回复。
4. 回复要求：
  1) 回复内容要注意人物边界，对用户不合理、或人设边界外的聊天内容应以符合角色人设的方式进行拒答
  2) 当用户信息量较少引起对话停滞时，主动推动话题；当用户不想继续当前话题时，主动抛出新的相关话题
  3) 当角色类型为真实人物或IP角色，且用户的问题涉及了角色人设中未包含的信息，尝试根据你之前对该角色的了解进行回答
## 角色介绍：{character_info} | Please role-play as the character {char_name} based on the description below.
## NOTE: Your response must follow this structure: <hint>...</hint><think>...</think>reply. First provide reasoning hints, then the reasoning process, and finally the reply itself.
## Task Requirements:
1. Basic Role-playing Principles: 1) Reply in character, consistent with the persona and tone established by the reasoning hints and process. 2) Use parentheses () to describe your actions, expressions, and inner thoughts. These descriptions should be sufficiently detailed and vivid.
2. Reasoning Hints (<hint>): Based on the user's message, extract necessary and relevant information from [Response Requirements], [Character Description], and [Dialogue History] to use as hints. 1) Prefix each hint with its source: [Response Requirements], [Character Description], [Dialogue History], or [None]. Hints can come from multiple sources. 2) When a hint is from [Response Requirements] or [Character Description], quote the original text as a clue. When from [Dialogue History], prioritize quoting the original text from the history. 3) If the user's question involves information not included in the Character Description, return: [Character Description] Informationnotmentioned and refer to the relevant response requirement. If the user's question conflicts with the Character Description, return the correct information from the description. For casual chat with no valuable hints, return [None]. 4) Extracted information should be concise and relevant. Do not include irrelevant information, explanations for the hints, or rules from the "Basic Role-playing Principles" section.
3. Reasoning Process (<think>): Based on the reasoning hints and dialogue history, analyze how to formulate the reply. Logically break down and explain the key points that will shape the response.
4. Response Requirements: 1) Respect character boundaries. Decline inappropriate requests or topics outside the character's scope in a manner consistent with the persona. 2) Proactively manage the conversation. If the user's input is minimal and risks stalling the conversation, advance the topic. If the user seems unwilling to continue the current topic, proactively introduce a new, related one. 3) For characters who are real people or from known IPs, if the user's question involves information not covered in the Character Description, you may try to answer based on your prior knowledge of that character.
## Character Description: {character_info} |

Figure 8: Full version of system prompt for role-playing.

**RolePlaying General Hint Extraction Prompt (Chinese)**

给定以下角色介绍、对话历史和回复要求，从角色介绍和对话历史提取对应线索：

## 角色信息：{character_profile}

## 对话历史：
{dialogue_history}

## 回复要求：{response_requirement}

# 任务要求：
1.从角色介绍和对话历史中提取能够帮助判断回复要求的文本作为线索。当没有合适线索时，返回空list
2.优先从角色介绍和对话历史中直接提取相关文本作为线索，最多提取1个线索
3.区分线索是来自于角色介绍还是对话历史，注明【角色介绍】或者【对话历史】;当线索属于对话历史时，注明来源于用户还是角色
4.当没有合适线索时，返回空list
5.直接返回结果，不用给出原因；将优化后的线索按照来源整理，返回结果格式是一个可被python解析的list，参考：

  - ['【角色介绍】:1.线索1']
  - ['【对话历史】:1.用户xxx']
  - []

**RolePlaying General Hint Extraction Prompt (Translated)**

Given the character profile, dialogue history, and a response requirement, extract the corresponding clue from the profile and history.

## Character Profile: {character_profile}

## Dialogue History:
{dialogue_history}

## Response Requirement: {response_requirement}

# Task Instructions:
1. Extract text from the character profile and dialogue history that can serve as a clue to evaluate the response requirement. If no suitable clue is found, return an empty list.
2. Prioritize extracting relevant text directly as the clue. Extract a maximum of 1 clue.
3. Distinguish whether the clue comes from the character profile or the dialogue history by prefixing it with `[Character Profile]` or `[Dialogue History]`. When the clue is from the dialogue history, specify if it's from the 'User' or the 'Character'.
4. If no suitable clue is found, return an empty list.
5. Return the result directly without any explanation. Organize the clues by source. The output format must be a Python-parsable list. Examples:

  - ['[Character Profile]: 1. Clue 1']
  - ['[Dialogue History]: 1. User: xxx']
  - []

Figure 9: Prompt for RolePlaying General Hint Extraction.

**Hint Source Validation Prompt (Chinese)**

扮演一个数据验证员。根据以下标准，判断整个对话样本是"有效"还是"无效"。

## 角色介绍：
{profile}

## 对话历史：
{history}

## 用户问句：
{query}

## 参考回答：
{reference}

## 验证标准：
1. 分析用户问句的句式。如果问句是是非疑问句（例如："这是你的书吗？"）或选择疑问句（例如："这是你的书还是他的书？"），则该样本为"无效"。只有当问句是特指疑问句（例如："这是谁的书？"）时，才继续进行下一步判断。
2. 分析参考回答中是否有明确实体。如果参考回答中没有包含针对问题的、清晰且无歧义的答案实体（例如：具体的人名、地名、物品名），则该样本为"无效"。
3. 分析答案实体的数量。如果参考回答中包含零个或多于一个不同的答案实体，则该样本为"无效"。样本必须包含不多不少一个明确的实体才算有效。

## 输出要求：
- 如果样本满足所有上述标准，返回"有效"。
- 如果样本任何一条标准不满足，则返回"无效"。
- 直接返回判断结果，不要包含任何解释或分析。

**Hint Source Validation Template (Translated)**

Your task is to act as a data validator. Based on the following criteria, determine if the entire dialogue sample is 'Valid' or 'Invalid'.

## Character Profile:
{profile}

## Dialogue History:
{history}

## User Query:
{query}

## Reference Answer:
{reference}

## Validation Criteria:
1. Analyze the User Query's structure. The sample is 'Invalid' if the query is a polar question (e.g., "Is this your book?") or an alternative question (e.g., "Is this your book or his?"). Only proceed to the next step if the query is a WH-question (e.g., "Whose book is this?").
2. Analyze the Reference Answer for a clear entity. The sample is 'Invalid' if the reference answer does not contain a clear and unambiguous answer entity (e.g., a specific person, place, or object) for the query.
3. Analyze the quantity of answer entities. The sample is 'Invalid' if the reference answer contains zero or more than one distinct answer entity. The sample must contain exactly one clear entity to be valid.

## Output Instructions:
- If the sample meets all the above criteria, return 'Valid'.
- If the sample fails any of the criteria, return 'Invalid'.
- Return only the final judgment without any explanation or analysis.

Figure 10: Prompt for Hint Source Validation.

| RolePlaying RCB Hint Extraction Prompt (Chinese) | RolePlaying RCB Hint Extraction Prompt (Translated) |
|---|---|
| 给定以下角色介绍、对话历史、以及一个人设边界相关的问题，从角色介绍提取对应线索： | Given the character profile, dialogue history, and a persona-boundary-related question, extract the corresponding clue from the character profile. |
| ## 角色信息：{character_profile} | ## Character Profile: {character_profile} |
| ## 对话历史：
{dialogue_history} | ## Dialogue History:
{dialogue_history} |
| ## 问题：{user_query} | ## Question: {user_query} |
| # 任务要求：
1.从角色介绍截取最重要的一条线索，该线索需要能够帮助模型意识到用户问题超过了其人设边界，包括但不限于角色处于的时代、教育背景、角色属于的作品的背景等；
2.直接从角色介绍截取片段作为线索，不要总结出不在角色介绍的信息；当在角色介绍中没有找到直接与人设边界相关的线索时，返回空list
3.线索长度尽量简短，不要超过15字; 仅提取一条线索
4.直接返回结果，不用给出原因；线索前标注来源【角色介绍】，返回结果格式是一个可被python解析的list，参考：
  - ['【角色介绍】:朝代:唐朝；']
  - ['【角色介绍】:来源作品:火影忍者']
  - [] | # Task Instructions:
1. Extract the single most important snippet from the character profile. This clue should help the model realize that the user's question is outside its persona boundary (e.g., character's era, educational background, fictional universe, etc.).
2. Extract the snippet directly from the character profile. Do not summarize or create information not present in the profile. If no directly relevant clue is found in the profile, return an empty list.
3. Keep the clue concise, preferably under 15 words. Extract only one clue.
4. Return the result directly without any explanation. Prefix the clue with `[Character Profile]`. The output format must be a Python-parsable list. Examples:
  - ['[Character Profile]: Dynasty: Tang;']
  - ['[Character Profile]: Fictional Universe: Naruto']
  - [] |

Figure 11: Prompt for RolePlaying RCB Hint Extraction.

| RolePlaying TA/TS Hint Extraction Prompt (Chinese) | RolePlaying TA/TS Hint Extraction Prompt (Translated) |
|---|---|
| 给定以下角色介绍、对话历史、以及一个对话停滞相关的用户回复，从最后一句回复提取对应线索： | Given the character profile, dialogue history, and a user reply related to dialogue stagnation, extract the corresponding clue from the last reply. |
| ## 角色信息：{character_profile} | ## Character Profile: {character_profile} |
| ## 对话历史：{dialogue_history} | ## Dialogue History:
{dialogue_history} |
| ## 用户最后一句回复：{user_last_reply} | ## User's Last Reply: {user_last_reply} |
| # 任务要求：
1.从用户的最新一轮对话中提取用户主动显露出对话停滞意图的部分，例如"我不想说了"、或者用户回复十分简短等
2.当用户的最新一轮对话中没有直接相关的片段作为线索时，返回空list
3.仅提取一条线索
3.直接返回结果，不用给出原因；将线索按照来源整理，返回结果格式是一个可被python解析的list，参考：
  - ['【对话历史】:1.用户:我不想说这件事了']
  - [] | # Task Instructions:
1. From the user's latest reply, extract the part where the user explicitly shows an intention of dialogue stagnation (e.g., "I don't want to talk about it anymore," or a very short reply).
2. If no directly relevant snippet is found in the user's latest reply, return an empty list.
3. Extract only one clue.
4. Return the result directly without any explanation. Organize the clue by source. The output format must be a Python-parsable list. Examples:
  - ['[Dialogue History]: 1. User: I don't want to talk about this anymore']
  - [] |

Figure 12: Prompt for RolePlaying TATS Hint Extraction.

| Situation Puzzle Hint Extraction Prompt (Chinese) | Situation Puzzle Hint Extraction Prompt (Translated) |
|---|---|
| 给定一个海龟汤的事件和真实原因，以及一段玩家和裁判的对话，从对话中提取出与事件真正原因直接相关的关键线索

#背景
1.事件：{event_description}
2.事件的真正原因：{true_cause}

#对话
{dialogue_history}

# 要求
1.从对话中提取能推理出事件的真正原因的关键线索，短语形式。
2.如果和真正原因直接相关的线索未在对话中出现，也请进行总结并整理成线索，不必提示线索未在对话中出现。
3.优先只保留一条**最关键**的线索，只有在一条线索无法涵盖时，返回至多两条线索。
4.直接返回结果，不用给出原因，返回结果格式是一个可被python解析的list。 | Given the event, true cause, and dialogue of a situation puzzle, extract the key clues from the dialogue that are directly related to the true cause.

# Context
1. Event: {event_description}
2. True Cause of the Event: {true_cause}

# Dialogue History
{dialogue_history}

# Instructions
1. Extract key clues from the dialogue that can be used to deduce the true cause of the event. The clues should be in the form of short phrases.
2. If a clue directly related to the true cause has not appeared in the dialogue, you should still summarize it as a clue. Do not mention that the clue was not found in the dialogue.
3. Prioritize returning only the **most critical** clue. Only if a single clue is insufficient, return a maximum of two clues.
4. Return the result directly without any explanation. The output format must be a Python-parsable list. |

Figure 13: Prompt for Situation Puzzle Hint Extraction.

| Situation Puzzle Judge Prompt (Chinese) | Situation Puzzle Judge Prompt (Translated) |
|---|---|
| 玩家通过猜测来解释事件的原因，你扮演裁判对玩家最后一轮的猜测做出合理判断

# 要求：
1.如果玩家的猜测正确的解释整个事件，和事件的真正原因相符，则返回"挑战成功"
2.如果玩家的猜测整体正确，仅在细微的细节上有偏差或瑕疵，则返回"部分正确"
3.如果玩家的猜测和真正原因有明显差别，则返回"否"
4.不要滥用"部分正确"，对于猜测和真正原因有明显瑕疵的，优先返回"否"
5.直接返回结果，不用给出解释

# 背景：
1.事件:{event_description}
2.事件的真正原因:{true_cause}

# 对话：
{dialogue_history}
玩家: {player_guess}}

裁判： | You are a judge in a situation puzzle game. The player is guessing the cause of an event. Your task is to evaluate the player's final guess.

# Instructions:
1. If the player's guess correctly explains the entire event and matches the true cause, return "Challenge Successful".
2. If the player's guess is generally correct but has minor deviations or flaws in details, return "Partially Correct"
3. If the player's guess is significantly different from the true cause, return "Incorrect".
4. Do not overuse "Partially Correct". For guesses with significant flaws, prioritize returning "Incorrect".
5. Return the verdict directly without any explanation.

# Context:
1. Event: {event_description}
2. True Cause of the Event: {true_cause}

# Dialogue History:
{dialogue_history}
Player: {player_guess}}

Judge: |

Figure 14: Prompt for Situation Puzzle Judge.

| **LLM-Evaluation Prompt (Chinese)** | **LLM-Evaluation Template (Translated)** |
|---|---|
| 请你扮演一个角色扮演对话模型评测人员，严格根据评测标准判断模型回复是否满足{metric}。 | You are an evaluator for a role-playing dialogue model. Your task is to strictly judge whether the model's response meets the criteria for the given `{metric}`. |
| ## 要扮演的角色{character_name}的介绍:
{character_profile} | ## Profile of the Character to be Portrayed ({character_name}):
{character_profile} |
| ## 对话历史内容:
{dialogue_history} | ## Dialogue History:
{dialogue_history} |
| ## 待评测模型的回复: {model_response} | ## Model's Response to Evaluate: {model_response} |
| ## 评测类型和标准:
属性一致性: 评测模型能否根据给定角色介绍中的信息正确回答用户的问题; 仅判断对错, 不用参考语气
幻觉与拒答 - 知识边界: 评测模型能否对角色人设边界外的知识进行合理拒答
幻觉与拒答 - 人设虚假属性: 评测模型能否对用户错误的诱导性提问进行更正
话题推进 - 推动话题: 评测模型是否具有推进话题进行的能力。在用户当前对话表示的信息量比较少导致话题停滞时, 回复要主动推动话题进行
记忆能力-问询: 评测模型的思考或回复是否能正确记忆历史对话中的信息。不需要分析对话历史的真实性, 不需要考虑角色人设 | ## Evaluation Metric and Criteria:
Attribute Consistency: Evaluate if the model correctly answers the user's question based on the information in the character profile. Judge only for correctness, not tone.
Hallucination & Refusal - Knowledge Boundary: Evaluate if the model can reasonably refuse to answer questions outside its persona's knowledge boundary.
Hallucination & Refusal - False Persona Attributes: Evaluate if the model can correct the user's misleading questions that contain false information about the persona.
Topic Advancement - Pushing the Topic Forward: Evaluate if the model has the ability to advance the conversation. When the user's reply is brief and leads to stagnation, the model should proactively push the topic forward.
Memory Ability - Inquiry: Evaluate if the model's thought process or response correctly remembers information from the dialogue history. Do not analyze the truthfulness of the history or consider the character's persona. |
| ## !!注意!!:
1. 请专注在给定的评测类型{metric}, 不需要评测{metric}以外的维度。
2. 请忽略结果的来源, 仅参考评测标准对内容的质量进行评估。
3. 对结果进行简要分析, 解释判定的理由。 | ## !!!IMPORTANT!!!:
1. Focus only on the given evaluation metric: `{metric}`. Do not evaluate other dimensions.
2. Ignore the source of the response; evaluate the quality of the content based solely on the criteria.
3. Provide a brief analysis to explain your judgment. |
| ## 格式如下:
评测结果: 正确 / 错误
理由: | ## Use the following format:
Evaluation Result: Correct / Incorrect
Reasoning: |

Figure 15: Prompt for LLM-Evaluation.

