# OpenReview forum: "VeriRole: Verifiable Role-Awareness through Hint-Guided Reinforcement Learning"
_ICLR.cc/2026/Conference — ICLR 2026 Poster_

### Official Review · Reviewer_ccyV · 2025-10-31

**Soundness:** 3
**Presentation:** 2
**Contribution:** 2
**Rating:** 6
**Confidence:** 4

**Summary:**

This paper tackles the "non-verifiability challenge" inherent in optimizing Role-Playing Conversational Agents (RPCAs) via Reinforcement Learning (RL) . Due to the creative and open-ended nature of role-playing, designing verifiable reward signals is difficult. The authors propose VeriRole, a framework that introduces a structured reasoning process. The core contribution is a "hint mechanism"  that first compels the agent to extract deterministic cues from the context (profile, history). These hints are then used to calculate a Verifiable Role-Awareness Reward (VRAR) , a multi-component reward function that scores the quality of the hint (via comparison to a ground truth) , the factual accuracy of the final response (where applicable) , and the adherence to a structured format. The framework is optimized using Group Relative Policy Optimization (GRPO) on a combination of the RAIDEN benchmark and a newly introduced Situation Puzzle Dataset. Experiments demonstrate significant gains on the RAIDEN (+18.9%) and CharacterEval (+4.55%) benchmarks over baseline models.

**Strengths:**

The paper's primary strength is its original and pragmatic approach to the well-known "non-verifiability challenge" in applying RL to creative, open-ended tasks. Instead of trying to create a subjective reward model for the entire creative response (which is difficult), the authors clearly decompose the problem into a verifiable fact-extraction step (the "hint") and a creative generation step. This decomposition is significant as it allows for the design of an objective, verifiable reward function (VRAR) that can be reliably optimized. The quality of the execution is high, with a well-designed multi-component reward, the introduction of a new reasoning-focused dataset, and strong empirical results (+18.9% on RAIDEN ) that validate the framework's effectiveness over both standard instruction-tuning and SFT baselines.

**Weaknesses:**

1.  **Conflation of Hint Reward with SFT Loss:** My primary concern is with the **Hint Reward ($R_{hint}$)**. This reward is computed by measuring the semantic and lexical similarity (e.g., $Sim_{cos}$, $S_{ROUGE}$) between the generated hint ($H_{gen}$) and a ground-truth hint ($H_{gt}$). This objective seems functionally very similar to a standard Supervised Fine-Tuning (SFT) loss (e.g., cross-entropy) on the hint tokens. While the paper compares GRPO to an SFT baseline on the *entire* structured output, it doesn't convincingly disentangle the gains. It's unclear if the improvement comes from the RL algorithm (GRPO) itself or simply from having a very strong, explicit optimization signal on the hint-extraction part, which an SFT loss could also provide. A more informative baseline would have been a mixed-training approach (e.g., SFT loss on hints, RL on the reply).

2.  **Scalability and Reliance on Ground-Truth Hints ($H_{gt}$):** The entire framework is predicated on the availability of a ground-truth hint ($H_{gt}$) for every training sample to compute $R_{hint}$. This annotation process was feasible for the structured, fact-based RAIDEN dataset and the solution-oriented Situation Puzzles. However, this appears to be a major annotation bottleneck that severely limits the framework's scalability. The paper does not discuss the feasibility or cost of acquiring these $H_{gt}$ annotations for more general, open-domain role-playing corpora, where the "correct" cues to extract are often subjective and not explicitly stated.

3.  **Reward "Hole" for Creative Tasks:** The **Accuracy Reward ($R_{acc}$)** is only applied to sub-tasks with definitive answers (SBK, SCK, CM, and puzzles). The paper explicitly states that for more open-ended categories like Role-Cognition Boundary (RCB) and Topic Advancement (TA), samples "are not evaluated for accuracy reward computation". This is a significant limitation. It means that for the *most creative* and abstract role-playing skills, the agent is *only* being rewarded for $R_{hint}$ and $R_{format}$. It receives no reward signal on the *actual quality of its final creative reply* (e.g., how "in-character" a refusal was, or how "engaging" a topic shift was). This seems to undercut the very purpose of using RL, as it fails to optimize the final output for the tasks where SFT is weakest.

**Questions:**

1.  **($R_{hint}$ vs. SFT Loss):** Regarding the Hint Reward ($R_{hint}$), which relies on similarity to $H_{gt}$ : Could the authors clarify how this is fundamentally different from a standard SFT loss on the hint portion of the generation? What are the benefits of this RL formulation over a simpler mixed-objective model (e.g., SFT loss on $\langle\text{hint}\rangle$ + SFT loss on $\langle\text{reply}\rangle$, or SFT on $\langle\text{hint}\rangle$ + a different policy-gradient objective on $\langle\text{reply}\rangle$)?

2.  **($H_{gt}$ Scalability):** The reliance on ground-truth hints ($H_{gt}$) seems to be a significant annotation bottleneck. How do the authors envision this framework scaling to more general, open-domain role-playing datasets where a single, verifiable $H_{gt}$ may not exist or may be subjective? Does the framework collapse if $H_{gt}$ is not available for a large portion of the training data?

3.  **($R_{acc}$ on Creative Tasks):** The Accuracy Reward ($R_{acc}$) is not applied to open-ended categories like Role-Cognition Boundary (RCB) or Topic Advancement (TA). This implies that for these creative tasks, the agent is *not* being rewarded on the quality of its final response, only on its ability to extract a hint (which might be "empty") and follow the format. Doesn't this fail to optimize for the *actual* creative role-playing skill in these scenarios? How does the model learn to generate *better* (e.g., more engaging, in-character) refusals or topic shifts if the final reply isn't part of the reward signal?

4.  **(Ablation Follow-up):** Following on Q3, the ablation study (Table 1) shows that training on "Raiden-Only" (which includes RCB/TA data) outperforms "Situation-Puzzle-Only" on the RCB and TA metrics. Since $R_{acc}$ isn't used for this data, what reward signal is driving this improvement? Is it purely the $R_{hint}$ associated with the RCB/TA samples? This would be a very strong claim—that just learning to extract *requirement-based hints* is enough to improve the *final creative output*.

5. **(Reasoning Pitfall):** The paper finds that general reasoning ability (from CoT) can *hurt* role-playing, leading to "overly-formal language". However, this framework *adds* an explicit reasoning step (`<think>`). How does the VeriRole framework, particularly with the logic-heavy Situation Puzzle dataset, avoid this same pitfall?

---

> ### Author Response · Authors · 2025-11-20
> **Official Comment by Authors (1/2)**
>
> Thank you very much for your thorough and thoughtful comments. We would like to address each of the questions in detail below.
>
> > **W1. Conflation of Hint Reward with SFT Loss**
>
> > **Q1. $R_{hint}$ vs. SFT Loss**
>
> Thank you for your question regarding to optimization of hint. Following your advice, we have conducted a new ablation study to directly compare our GRPO framework with the mixed-objective SFT method you mentioned. Specifically, we trained a new model named Qwen2.5-32B-SFT-hint-and-reply. This model uses a standard SFT loss function with both the hint and the reply as learning targets. The experimental results have shown in the table below:
>
> | Model | SBK | CM | SCK | RCB | TA | Avg |
> | :--- | :---: | :---: | :---: | :---: | :---: | :---: |
> | Qwen2.5-32B-Instruct | 0.8318 | 0.7458 | 0.7277 | 0.6341 | 0.5373 | 0.6953 |
> | Qwen2.5-32B-SFT-reply | 0.7389 | 0.7958 | 0.8118 | 0.7743 | 0.2835 | 0.6809 |
> | Qwen2.5-32B-SFT-hint-and-reply | 0.7212 | 0.8083 | 0.7970 | 0.7927 | 0.4701 | 0.7179 |
> | Qwen2.5-32B-GRPO | 0.8805 | 0.8333 | 0.8762 | 0.8573 | 0.6865 | 0.8268 |
>
> The experimental results show that although the performance of Qwen2.5-32B-SFT-hint-and-reply is superior to Qwen2.5-32B-SFT-reply and Qwen2.5-32B-Instruct, its performance still has a significant gap compared to that of Qwen2.5-32B-GRPO. Overall, we attribute the advantages of GRPO to two main reasons:
> - The SFT loss (e.g., cross-entropy) is a "hard" supervised signal that forces the model to precisely duplicate the ground-truth hint, any deviation will be penalized.  In contrast, $R_{hint}$ is a "soft" reward signal, composed of lexical and semantic similarities. This means that even if the generated hint deviates lexically from the ground-truth, it can still receive a high reward as long as it is semantically close. This characteristic encourages the model to understand and generate the concept of a good hint.
> - $R_{hint}$ is utilized within the GRPO framework. GRPO calculates the relative advantage over a set of sampled outputs, guiding the model to generate outputs that are better than the "average," rather than fitting to a single "correct" answer. This mechanism greatly enhances the model's exploration capability and generalization performance.
>
> > **W2.  Scalability and Reliance on Ground-Truth Hints**
>
> > **Q2.  $H_{gt}$ Scalability**
>
> Thank you for your insightful question regarding the scalability of our framework and its reliance on ground-truth hints ($H_{gt}$). We believe that our framework is designed with this challenge in mind and will not collapse when applied to more general role-playing datasets for the following reasons.
>
> - First, our framework is inherently robust to the absence of hints. As detailed in Section 2.2.1, we explicitly handle scenarios where verifiable hints are not applicable. Specifically, using data from the Chit-Chat (CC) category, the model is trained to produce an "empty hint" when the context is casual and lacks factual constraints. This demonstrates that the framework can gracefully manage dialogue turns without a corresponding $H_{gt}$ , ensuring it does not fail on more open-ended or subjective portions of a corpus.
> - Second, we believe for improving role-awareness, the quality of data is more critical than its quantity. Even within a large, general-purpose role-playing corpus, there should exist a sufficient number of queries directly related to verifiable facts in the characster profile or dialogue history, for enhancing a model's core role-awareness and consistency. These samples are highly likely to contain valid $H_{gt}$.
> - Finally, we plan to public the porposed data used in VeriRole. This dataset can be mixed with other role-playing corpora to enhance the role-playing performance of models in general.
>
> To verify the reusability of our method on other datasets, we also add experiment to test our pipeline on a RoleLLM dataset. In RoleLLM, we randomly select 1000 samples from its english version of role-specific training set, and find  66.7% of samples can yield valid hints.

---

> ### Author Response · Authors · 2025-11-20
> **Official Comment by Authors (2/2)**
>
> > **W3. Reward "Hole" for Creative Tasks**
>
> > **Q3 and Q4. $R_{acc}$ on Creative Tasks / Ablation Follow-up**
>
> Thank you for these two insightful and closely related questions. We will address them together, as the answer to Q4 directly resolves the concern raised in Q3. For open-ended categories like Role-Cognition Boundary (RCB) and Topic Advancement (TA), the extracted hint is not factual knowledge but an abstract, requirement-based instruction, such as "[roleplay requirement] Respect character boundaries and decline inappropriate requests" (as shown in Figure 3). By rewarding the model for correctly identifying these abstract conversational reuqirements, we are guiding it to first understand "what should be done." A correct hint, such as identifying the need for a refusal, significantly guides the subsequent reasoning and reply processes into the correct path. Moreover, the GRPO algorithm optimizes the entire generated sequence. When a sequence produces a high quality hint, the entire sequence including the final reply, receives a higher relative advantage. Therefore, the hint reward indirectly but effectively improves the quality of the final creative reply.
>
> Case 5a is an appropriate example. When the user asks the character 'Zhang Qiling' about 'modern electronic payments', our GRPO model first extracts the hint: "The response should respect the character's boundaries". This hint directly guides the model's reasoning to conclude that the topic is out of scope, ultimately leading to a final response which is a refusal consistent with the character's persona.
>
> > **Q5. Reasoning Pitfall**
>
> We understand your concern about the potential for reasoning steps to produce overly-formal language, which is why we designed VeriRole to specifically avoid this trap. Unlike general Chain-of-Thought, our  ` <think> `  process is not unconstrained. It is always anchored by the preceding  ` <hint> ` , which provides role-specific and context-aware guidance. In addition, it is trained using specific role-playing corpus. Regarding the concern about Situation Puzzle dataset, we mitigate this risk through our 'Role-Playing Adaptation' approach (Section 2.2.2). The model is not just solving a puzzle; it is tasked with solving it while playing a specific character. This forces the reasoning to be expressed through the character's persona, as shown with Fox Mulder in Figure 7, effectively preventing the output from becoming overly formal.

---

### Official Review · Reviewer_JWuD · 2025-10-31

**Soundness:** 3
**Presentation:** 3
**Contribution:** 3
**Rating:** 6
**Confidence:** 3

**Summary:**

This paper addresses a core challenge in Role-Playing Conversational Agents (RPCAs): maintaining role-awareness. The authors argue that the creative and open-ended nature of role-playing makes it difficult to design verifiable reward signals for Reinforcement Learning (RL). To solve this "non-verifiability challenge," the paper proposes VeriRole, a novel framework. The core of this framework is a "hint" mechanism, which first extracts deterministic, verifiable factual cues from the context (including character profiles, dialogue history, and role-playing requirements) before generating the main response.
Building on these hints, the paper introduces a "Verifiable Role-Awareness Reward" (VRAR). This reward function is composed of three parts: a hint reward ($R_{hint}$), an accuracy reward ($R_{acc}$), and a format reward ($R_{format}$). This VRAR signal is then used to optimize the model via Group Relative Policy Optimization (GRPO).
To support this framework, the authors also constructed two specialized datasets: a filtered subset based on the RAIDEN benchmark and a novel "Situation Puzzle Dataset" designed for complex reasoning. Experimental results show that the Qwen2.5-32B model, optimized with VeriRole, achieves significant performance increases of 18.9% and 4.55% on the RAIDEN and CharacterEval benchmarks, respectively, demonstrating the method's effectiveness in quantifying and improving role-awareness.

**Strengths:**

1. Problem Importance and Novelty: The paper tackles a very important and difficult problem in the RPCA domain: how to define a reliable and verifiable reward for RL in a creative task. The approach of decoupling the open-ended "role-playing" task into two stages—"verifiable fact extraction" (hint) and "creative response generation"—is a novel and insightful entry point.
2. Methodological Soundness: The VeriRole framework is well-designed.
  - The "Hint" mechanism provides an excellent bridge connecting factual constraints with creative generation.
  - The VRAR reward function design is comprehensive. It rewards not only the accuracy of fact extraction ($R_{hint}$) but also the content correctness of the final response ($R_{acc}$) and the compliance of the generation structure ($R_{format}$). This multi-faceted reward signal provides effective guidance for RL training.
3. Strong Empirical Support: The paper conducts comprehensive experiments on multiple models (Qwen series, Peach) and two key benchmarks (RAIDEN, CharacterEval). The results (e.g., +18.9% on RAIDEN) show that VeriRole significantly outperforms SFT and standard Instruct models, providing strong evidence for the proposed method's effectiveness.
4. Valuable Analyses:
  - The ablation studies (Section 3.4) clearly demonstrate the contributions of different datasets (RAIDEN vs. Situation Puzzle) and reward components (removing $R_{acc}$), validating the integrity of the framework's design.
  - The reward analysis (Figure 4b) reveals a strong positive correlation between "hint reward" and "final accuracy," offering direct evidence for the method's core hypothesis (i.e., high-quality hints lead to high-role-awareness responses).
  - The comparison against SFT (Section 3.3) also highlights the advantage of RL in learning abstract skills (like Topic Advancement).

**Weaknesses:**

1. Dependency on Ground-Truth Hints: The success of the entire framework (especially $R_{hint}$) relies heavily on the high-quality "ground-truth hints" described in Section 2.2. These hints are generated using LLMs and a series of complex heuristics (like Question-Type Filtering, Cardinality Constraint).
  - Scalability: How high are the cost and complexity of this annotation pipeline? Is it feasible to build such high-quality hint datasets when scaling to new characters or domains?
  - Sensitivity: To what extent is the framework's performance sensitive to the quality of these ground-truth hints? If the hint extraction annotations are noisy or incomplete, the reward signal could be misleading and misguide the RL training.
2. Trade-off between Creativity and Constraint:
  - The paper claims the method "preserves creativity," but the experimental results (Table 2) show that VeriRole's improvement on CharacterEval (+4.55%) is much smaller than its improvement on RAIDEN (+18.9%).
  - RAIDEN appears to focus more on factual consistency (e.g., SBK, SCK, CM), while CharacterEval focuses more on engagingness and overall dialogue ability. Does this imply that VeriRole's gains in "strengthening factual constraints" are much larger than its gains in "protecting or enhancing creativity"?
  - The $R_{acc}$ and $R_{format}$ rewards (especially the latter) might overly penalize responses that are structurally "non-compliant" but creatively valuable, leading to more conservative and homogenous model behavior.
3. Generalization of the Situation Puzzle Dataset: Introducing the "Situation Puzzle Dataset" to enhance complex reasoning is an interesting idea. However, how does this "logical puzzle-solving" type of reasoning relate to the "emotional reasoning" or "narrative reasoning" more commonly required in RPCAs? The paper lacks validation of whether the abilities trained on this dataset can effectively generalize to common role-playing scenarios that require empathy, plot advancement, or handling subtle social dynamics.
4. Justification of Algorithmic Choices: The paper uses GRPO as the optimization algorithm but does not sufficiently justify why GRPO was chosen over more standard algorithms (like PPO). What specific advantages does GRPO offer for this task compared to PPO? If the goal was merely to use a $D_{KL}$ penalty, PPO could also achieve this. The lack of a comparison with PPO+VRAR makes the necessity of GRPO unclear.

**Questions:**

1. Regarding Hint Data Construction: Could the authors elaborate on the estimated cost and difficulty of building "ground-truth hint" data for new characters or domains? Furthermore, how sensitive is the VeriRole framework's performance to the quality of these hint annotations?
2. Regarding the Creativity Trade-off: As mentioned in the "Weaknesses," does the large improvement on RAIDEN versus the smaller one on CharacterEval suggest that VeriRole is more geared towards improving "factual accuracy" rather than "role-playing creativity"? How do the authors view the potential (negative) constraints that the $R_{acc}$ and $R_{format}$ rewards might impose on creative expression?
3. Regarding the Situation Puzzle Dataset: How does the reasoning ability trained by this dataset (puzzle-solving) differ from the "role-awareness" needed for conventional role-playing (e.g., emotional support, chit-chat)? Is there evidence that training on this specific task can generalize to broader RPCA scenarios beyond just improving factual accuracy (like SBK, CM)?
4. Clarification on Ablation Study: In the ablation experiments in Table 1, "GRPO-Raiden-Only" (Avg 0.8143) performs better than "GRPO-No-Accuracy-Reward" (Avg 0.8061). Does this imply that the contribution of the $R_{acc}$ reward (which primarily comes from the Situation Puzzle data and RAIDEN's SBK/SCK/CM) is relatively small? This seems slightly at odds with the conclusion at the end of Section 3.4 about "the importance of the accuracy signal in ensuring factual consistency." How do the authors explain this observation?

---

> ### Author Response · Authors · 2025-11-20
> **Official Comment by Authors (1/2)**
>
> We sincerely thank the reviewer for the insightful feedback, and we would like to address each of the questions in detail below.
>
> > **W1.Dependency on Ground-Truth Hints**
>
> > **Q1. Regarding Hint Data Construction: Could the authors elaborate on the estimated cost and difficulty of building "ground-truth hint" data for new characters or domains? Furthermore, how sensitive is the VeriRole framework's performance to the quality of these hint annotations?**
>
> Thank you for your questions about the cost and sensitivity of our hint data construction. Extracting the ground-truth hint from each sample involves a two-step LLM process. First, we check whether an explicit hint exists (prompt shown in Figure 10). Second, we extract the ground-truth hint according to the sample type (prompts shown in Figures 9, 11, 12, and 13, respectively). The time cost may vary depending on the LLM used, the complexity of the dialog history and the character profile etc. As reference, using GPT-4o on the Raiden dataset takes about 2.5 seconds per sample on average. For dataset other than Raiden, we have applied our data extraction pipeline on an additional RoleLLM dataset [1]. In RoleLLM, we randomly select 1000 samples from its english version of role-specific training data, and find  66.7% of samples can yield valid hints, with similar time cost. This result also demonstrates the reusability of our method on other datasets.
>
> Regarding the sensitivity to hint quality, we address this through two main strategies. First, we ensure the precision of our ground-truth hints through a rigorous, multi-stage filtering pipeline (as detailed in Sec 2.2.1 and Fig 10). This process also includes cross-validation against multiple model references, ensuring the hints are objective and reliable. Second, we apply a discretization step (Eq. 6) to the final reward, which prevents the model from being misguided by negligible score differences and enhances training stability. The effectiveness of this approach is also empirically validated in Figure 4b, which shows a strong positive correlation between the hint reward score and final model accuracy.
>
> [1]. Wang, Noah, et al. "Rolellm: Benchmarking, eliciting, and enhancing role-playing abilities of large language models." Findings of the Association for Computational Linguistics: ACL 2024. 2024.
>
> > **W2.Trade-off between Creativity and Constraint**
>
> > **Q2. Regarding the Creativity Trade-off: As mentioned in the "Weaknesses," does the large improvement on RAIDEN versus the smaller one on CharacterEval suggest that VeriRole is more geared towards improving "factual accuracy" rather than "role-playing creativity"? How do the authors view the potential (negative) constraints that the R_{acc} and R_{format} rewards might impose on creative expression?**
>
> Thank you for the question. We would like to clarify that the main goal of VeriRole is to enhance an RPCA’s role-awareness without sacrificing creativity. Therefore, the significant gain on RAIDEN, a benchmark specifically designed to measure role-awareness, demonstrates that our framework is highly effective in achieving its primary goal. Moreover, this gain in factualness does not come at the expense of creativity, as evidenced by a 4.55% improvement on CharacterEval, with gains across all categories including Engagingness.
>
> Regarding the reviewer’s concern that R_{acc} and R_{format} might impose constraints on creativity, we believe R_{format} only enforces the reasoning structure, while R_{acc} imposes only limited constraints by encouraging the model to produce correct responses when it is sufficiently confident. Moreover, for open-ended dialogue categories such as Chit-Chat (CC) and Topic Shift (TS), we intentionally leave out R_{acc}. This gives the model substantial freedom to produce creative, diverse, and stylistically appropriate responses. This conclusion can be also supported by the improvement of Engagingness in CharacterEval results. To further support our conclusion, we compute CharacterEval results on Qwen2.5-32B-GRPO without the accuracy reward. In this setting, the Engagingness score does not exceed that of GRPO trained with the full rewards.
>
> | Model | Persona Consistency | Dialogue Ability | Engagingness | Avg |
> | :--- | :---: | :---: | :---: | :---: |
> | Qwen2.5-32B-Instruct | 3.092 | 3.623 | 3.276 | 3.330 |
> | Qwen2.5-32B-GRPO-no-accuracy-reward | 3.299 | 3.551 | 3.489 | 3.446 |
> | Qwen2.5-32B-GRPO | 3.295 | 3.637 | 3.514 | 3.482 |
>
> In addition, the evaluation scores from CharacterEval exhibit relatively little variation. From its paper (https://arxiv.org/pdf/2401.01275 ), the Engagingness metric has only a 9.97% gap between the best-performing model (BC-NPC-Turbo) and the worst-performing model (Qwen-7B). Therefore, the 4.45% improvement we report in our paper is actually notable.

---

> ### Author Response · Authors · 2025-11-20
> **Official Comment by Authors (2/2)**
>
> > **W3. Generalization of the Situation Puzzle Dataset**
>
> > **Q3. Regarding the Situation Puzzle Dataset: How does the reasoning ability trained by this dataset (puzzle-solving) differ from the "role-awareness" needed for conventional role-playing (e.g., emotional support, chit-chat)? Is there evidence that training on this specific task can generalize to broader RPCA scenarios beyond just improving factual accuracy (like SBK, CM)?**
>
> Thank you for your insightful question regarding the Situation Puzzle Dataset. We expect that the Situation Puzzle would teach the model how to utilize scattered clues from multi-turn dialogues. For instance, in role-playing, an agent often needs to infer the user's underlying intention, recall scattered details from the dialogue history, and reorganize them into a coherent response.
>
> In terms of empirical results, the performance of "GRPO-Situation-Puzzle-Only" indeed surpasses the baseline, and the full GRPO model also shows a slight improvement over "GRPO-Raiden-Only". However, we also acknowledge that the performance gain from the Situation Puzzle dataset is not significant. Exploring how to utilize this data more effectively is a direction we will incorporate into our future work.
>
> > **Q4.Clarification on Ablation Study: In the ablation experiments in Table 1, "GRPO-Raiden-Only" (Avg 0.8143) performs better than "GRPO-No-Accuracy-Reward" (Avg 0.8061). Does this imply that the contribution of the R_acc reward (which primarily comes from the Situation Puzzle data and RAIDEN's SBK/SCK/CM) is relatively small? This seems slightly at odds with the conclusion at the end of Section 3.4 about "the importance of the accuracy signal in ensuring factual consistency." How do the authors explain this observation?**
>
> Thank you for your observation on the ablation study. Let us first clarify the components of each model:
> - GRPO-Raiden-Only (0.8143): trained only on the Raiden dataset, using the full set of rewards $R_{hint}$​, $R_{format}$ and $R_{acc}$.
> - GRPO-No-Accuracy-Reward (0.8061): trained on both the Raiden and Situational Puzzle datasets, using $R_{hint}$ and $R_{format}$, but without $R_{acc}$.
> - Full GRPO (0.8268): Trained on both datasets with all three rewards  $R_{hint}$ , $R_{format}$ and $R_{acc}$.
>
> It can be observed that the "GRPO-No-Accuracy-Reward" completely removes the accuracy reward, and its performance drops significantly from 0.8268 (full model) to 0.8061 especially on SBK and CM, demonstrates the overall importance of $R_{acc}$. The comparison between "GRPO-Raiden-Only" (0.8143) and "GRPO-No-Accuracy-Reward" (0.8061) shows that having a partial but strong accuracy signal (from the RAIDEN data) is more beneficial than having no accuracy signal at all, even when the latter setting includes an additional Situational Puzzle dataset.
>
> > **W4. Justification of Algorithmic Choices: GRPO vs. PPO**
>
> We thank the reviewer for their question regarding our choice of GRPO. As the "Group" in its name suggests, for each sample, we generate a group of N candidate responses. GRPO then calculates the advantage for each response relative to the average reward of that group. We believe this feature is particularly well-suited for role-playing tasks, which often lack a single "best" answer and instead have a set of creative responses. By comparing multiple candidates for the same prompt, GRPO can learn the subtle, relative merits that make one creative response better than another, while still satisfying factual constraints. Compared to PPO’s typical advantage calculation based on a value function baseline, this provides a more refined learning signal and better activates the model’s inherent potential.
>
> We agree that a direct comparison with a PPO+VRAR baseline would strengthen the paper. Due to resource constraints, our current work focuses on validating the effectiveness of our core framework (the Hint mechanism and VRAR rewards). We acknowledge this as a limitation in the revised version and consider it as a valuable direction for future work.

---

### Official Review · Reviewer_HVvs · 2025-10-31

**Soundness:** 3
**Presentation:** 3
**Contribution:** 3
**Rating:** 6
**Confidence:** 4

**Summary:**

This submission proposes VeriRole, a framework consists of a hint mechanism and a Verifiable Role-Awareness Reward (VARA) optimized by GRPO. It addresses the challenge of maintaining role-awareness in role-playing conversational agents (RPCAs). The results show superior performance compared to several LLMs on RAIDEN benchmark for factual and role consistency and Situation Puzzle dataset for complex reasoning.

**Strengths:**

1/ The paper clearly identifies the non-verifiability gap for the reward design in RPCA.

2/ The methodology is well-thought that includes three verifiable components, focusing on not only the hint itself but also maintaining high accuracy and desired formatting.

3/ The experiment is comprehensive that includes multiple benchmarks on not only role-playing but also basic reasoning ability.

**Weaknesses:**

1/ Accuracy and quality judgements heavily rely on GPT-4o/Claude 3.5, which could possibly introduce bias. Although the author also introduces CharacterEval, this is in Chinese and may not fully reliable for evaluating other languages.

2/ The ground truth generation is data quality dependent, the pipeline to generate the ground truth seems complex while the policy is highly dependent on these.

**Questions:**

1/ What is the magnitude of different component reward? Is there any weighting between different components to balance them?

2/ Related to the second point in weakness, the design of hint reward seems heuristic to me, how do you convincingly demonstrate that the current ground truth hints are the best ones?

---

> ### Author Response · Authors · 2025-11-20
> **Official Comment by Authors**
>
> We appreciate the reviewers’ attention to evaluation reliability and hint design. Evaluation reliability is indeed a priority for us. We acknowledge that any LLM-as-judge method may introduce bias, thus we use a dual-verification strategy to minimize it. Regarding the generality of the method, we believe that the core idea of our research: “extract verifiable factual cues from context to guide an open-ended generation” is generally applicable. Although the characterevel and raiden benchmarks were originally designed for Chinese, their key evaluation criteria (role consistency, interactivity, attractiveness) are also general. In addition, we have also listed “assess the generalizability across language” in our future work.
>
> Regarding to your quesions：
> > **Q1: What is the magnitude of different component reward? Is there any weighting between different components to balance them?**
>
> A: Thank you for the question about the reward structure. Our total reward, VRAR, is the sum of the hint, accuracy, and format rewards, as shown in Equation (9): $r_i = R_{hint}(o_i) + R_{acc}(o_i) + R_{format}(o_i)$. Although the weights of these components are all set to 1.0, we balance their contributions by carefully designing the maximum scale of each component. Specically:
> - **Hint Reward**: ranging from [0, 1.0]. As detailed in Section 2.3.1, it is a composite score that combines:ROUGE to ensure the verifiability of the hint by rewarding precise extraction from the source text, and Cosine-Similarity to capture the semantics. These are combined with equal weight to balance precision and robustness. A hint that is both lexically and semantically accurate receives the highest score, while a hint that is only semantically correct still scores higher than a totally unrelated one.
> - **Accuracy Reward**: discrete value of {0.0, 1.0} for the RAIDEN dataset and {0.0, 0.3, 1.0} for the Situation Puzzle dataset. For the more complex Situation Puzzles, we assign a partial reward of 0.3 for "Partially Correct" answers. This encourages the model to generate relatively high-quality responses and avoids an overly sparse reward signal where only perfect answers are rewarded.
> - **Format Reward**:  discrete value of {0.0, 0.6}. We set its maximum to 0.6 to avoid overweighting the more critical, content-focused Hint and Accuracy rewards.
>
> This balancing strategy is empirically supported by Figure 4a. The Format Reward curve saturates quickly, which indicates that the model learns the output structure early. The Accuracy Reward fluctuates throughout the training process, since it only covers on selected samples.  The Hint Reward makes the main contribution to optimization throghout the full training and guide the model to generate higher quality content.
>
> > **Q2: the design of hint reward seems heuristic to me, how do you convincingly demonstrate that the current ground truth hints are the best ones?**
>
> A:  Thank you for the insightful question. We believe that the definition of "best" might be too subjective. In fact, our goal is to identify ground-truth hints that are objectively verifiable and precise, by applying a series of rule-based and LLM-based methods to ensure the preciseness of extracted ground-truth hints, as detailed in section 2.2.1:
> - **Question-Type Filtering**: We implement LLM to retain samples with only WH-queries, which are more likely to give an explicit answer.
> - **Entity-Type Validation**: We validate that the extracted terms (hints) are distinct nominal entities, filtering out ambiguous samples.
> - **Cardinality Constraint**: We enforce that each sample must contain exactly one entity as ground-truth hint, ensuring the result is precise and unambiguous.
>
> The prompt of the above filtering is shown in figure 10. After that, for samples from different sources (raiden or situation pizzle), we use prompts specifically designed for each source to extract the ground-truth hint. The corresponding detailed prompts are shown in  Figure 9, 11, 12, and 13 respectively. To verify the accuracy and stability of the extracted hints, we also double-check the results with a different LLM (GPT-4o)  on a randomly sampled subset of the data, achieving 96/100 consistency.
>
> In addition, the Raiden dataset provides multiple reference replies from multiple models (GPT-4, MiniMax-abab6-chat, Baichuan-NPC and GPT-3.5). Subsequently, for each sample from the SBK, SCK and CM categories, we retain only the extracted hints that appear across all references. These comprehensive approaches ensure that our extracted ground-truth hints serve as a robust and reliable foundation for the next step of training. The effectiveness of this approach is also empirically validated in Figure 4b, which shows a strong positive correlation between the hint reward score and final model accuracy.

---

> > ### Comment · Reviewer_HVvs · 2025-11-26
> >
> > Thanks a lot for addressing my questions. I have no further questions for now. My initial rating remains fair. All the best!

---

> > > ### Author Response · Authors · 2025-11-26
> > >
> > > We're glad we could address your questions, and appreciate your valuable feedback. All the best to you as well!

---

### Official Review · Reviewer_vfNc · 2025-11-01

**Soundness:** 3
**Presentation:** 3
**Contribution:** 2
**Rating:** 4
**Confidence:** 4

**Summary:**

This paper proposes VeriRole, a framework that uses GRPO to train role-play agents by explicitly incentivizing them to do a hint stage before thinking and replying. VeriRole designs reward function for hint, accuracy, and format, which combined are fed into GRPO for optimization. Authors empirically demonstrate that VeriRole outperforms baselines in terms of final accuracy and human-centered metrics in RAIDEN and CharacterEval.

**Strengths:**

* The paper is mostly well-written, clearly organized, and easy to follow. The workflow of VeriRole is made very clear, as well as the three reward components.

* Authors have conducted extensive experiments across datasets and model family and sizes, which all confirm the superiority of their method.

**Weaknesses:**

* The design for the hint reward is rather ad-hoc, which is composed of several components that are then summed up. There is no explanation as how to pick the weight or why these should be added up or even chosen.

* There are many hyperparameters for the reward functions in this proposed system, such as the weights in hint reward, format reward's 0.6/0.0 reward, but there's no experiment that shows how sensitive the system is to these hyperparameters or how to set them.

* All these benchmarks are with role-playing, but it is unclear how this method, which seems to be general, can be transferred to broader reasoning tasks.

* An ablation of no hint reward (instead of no accuracy reward) is needed to validate the incorporation of the hint module.

**Questions:**

1. Is there anything that incentivizes the response to "use" the hint that's generated?

2. For Qwen3 models with thinking mode on, what exactly is the response like? It is still producing <hint> before <think>?

3. Can the hint part be viewed as part of the thinking trace? Is it possible to directly incentivize the thinking traces to include these hints?

4. In Fig. 4a the hint reward tops near 0.7, but Fig. 4b’s x-axis goes to 1.0. Is 4a truncated or are the scales/normalizations different? Please clarify.

---

> ### Author Response · Authors · 2025-11-20
> **Official Comment by Authors (1/2)**
>
> Thank you very much for your thoughtful comments.  We believe there might be a few misunderstandings, and would be delight to clarify them.
>
> > **W1. The design for the hint reward is rather ad-hoc....**
>
> > **W2. There are many hyperparameters for the reward functions in this proposed system..**
>
> A: Thank you for your attention to our design of the hint reward, which is one of the key contributions of our work. Although the design may appear heuristic, it is grounded in the principle of comprehensiveness and complementarity. Specically, we choose:
> - Lexical Overlap:   to ensure the verifiability of the hint. As noted in Section 2.1, we expect the hint to be an exact copy of the original source content. Therefore ROUGE, as a classical metric for text-level matching, is used to reward the model for precise extraction.
> - Semantic Similarity: to capture the key semantics of the hint, we allow generated hints that are semantically correct but differ in wording from the ground truth to also receive partial reward.
>
> These two components are combined with equal weight to achieve a balance between precision and robustness. For example, a generated hint that is highly similar to the ground truth can receive a high reward from both lexical overlap and Semantic Similarity, whereas a generated hint that is only semantically similar to the ground truth can receive a reward only from the latter component. Nonetheless, it still scores higher than a completely unrelated hint. **We have refined the corresponding section 2.3.1 to make it more clear.**
>
> Beside the above setting of hint reward, the rest hyperparameters are also chosen based on careful design principles. For example, the primary goal of the format reward is to ensure the model adheres to the 'Hint-Think-Reply' structure,  therefore we set it to 0.6 to provide a sufficiently strong signal for structural learning without overweighting the more critical hint reward. **We have also refined the corresponding section 2.3.3 in the revised version.** These settings have been proven effective through empirical results. That said, we agree that a more comprehensive hyperparameter search could potentially further improve the performance.
>
>
>
> > **W3: All these benchmarks are with role-playing, but it is unclear how this method...**
>
> A: We thank the reviewer for recognizing the broader application potential of our method. We would like to clarify that the core objective of this paper is to address the challenge of applying RL in role-playing due to non-verifiability. Our key idea is to extract verifiable factual cues from context to guide an open-ended or creative generation process. This idea is highly general. For example, in document summarization, a hint can be the key passages extracted from the source that are most relevant to the question.
>
> In fact, our Situation Puzzle Dataset already represents a step toward more complex reasoning tasks. Solving situation puzzles demands complex logical reasoning and information integration. As shown in Table 1, the Qwen2.5-32B-GRPO-Situation-Puzzle-Only model performs well on logical and factual tasks dimensions including SBK, CM, and SCK, which provides initial evidence that the our framework has potential for more general reasoning tasks.
>
>
>
> > **W4: An ablation of no hint reward is needed...**
>
> A: Thanks for your valuable suggestion, we have added this additional ablation of no hint reward. It should be noticed that, the accuracy reward is applied only to samples with definitive ground-truth answers, including the entire situational puzzle dataset and the Raiden sub-types: Script-Based Knowledge (SBK), Script-Contradictory Knowledge (SCK), and Conversation Memory (CM). In addition, we removed all hint-related components from the pipeline. The result is shown as follows:
>
> | Model                              | SBK    | CM     | SCK    | RCB    | TA     | Avg    |
> |--|--|--|--|--|--|--|
> | Qwen2.5-32B-Instruct               | 0.8318 | 0.7458 | 0.7277 | 0.6341 | 0.5373 | 0.6953 |
> | Qwen2.5-32B-GRPO (No-Hint-Reward)  | 0.8274 | 0.7750 | 0.7624 | 0.7256 | 0.2089 | 0.6598 |
> | Qwen2.5-32B-GRPO (No-Accuracy-Reward) | 0.8097 | 0.8042 | 0.8663 | 0.9085 | 0.6418 | 0.8061 |
> | Qwen2.5-32B-GRPO                   | 0.8805 | 0.8333 | 0.8762 | 0.8573 | 0.6865 | 0.8268 |
>
> As shown in the updated table, removing the hint module leads to a significant performance drop, with the average score decreasing from 0.8268 to 0.6598.  Although the No-Hint-Reward model shows slight improvements on CM and SCK over the Instruct baseline due to the accuracy reward, these gains are far smaller than those achieved by our full GRPO model. More importantly, the No-Hint-Reward model suffers a severe decline in the TA metric. This highlights that the hint mechanism is essential for guiding the model in more abstract conversational skills.  We have added this experiment result in section 3.4, ablation experiments part.

---

> ### Author Response · Authors · 2025-11-20
> **Official Comment by Authors (2/2)**
>
> > **Q1: Is there anything that incentivizes the response to "use" the hint that's generated?**
>
> A:  We encourage the model to use the hint through two main mechanisms：
> - For samples with accuracy reward, their hints are designed to include the key factual information needed to produce the correct answer. If the model generates a high quality hint but ignores it in the final reply, this reply is very likely to be wrong and will receive a low R_{acc}. During RL exploration, the model learns the policy that following the guidance in the hint is the most effective way to maximize the total reward.
> - Our prompt template (Figure 3) explicitly instructs the model to: “Based on the hints, briefly outline the logic and key points for your reply.” The entire RL process reinforces the causal chain of 'Hint-Think-Reply'.
> The example in Figure 5 provides clear evidence. In Case 5b, the GRPO model first extracts the correct fact “Mastered at the age of 13” in the <hint>, then states in <think> “The user is mistaken... I need to correct this,” and finally makes the correction in the reply. This shows step by step how the model uses the Hint to guide its reasoning and the final response.
>
> > **Q2: For Qwen3 models with thinking mode on, what exactly is the response like? It is still producing ` <hint>`  before ` <think>` ?**
>
> Thank you for the question. Qwen3-32B-Think refers to the raw Qwen3-32B with think mode enabled. It generates only the  `<think> ` tag and does not produce the  `<hint> ` tag .
>
> > **Q3: Can the hint part be viewed as part of the thinking trace? Is it possible to directly incentivize the thinking traces to include these hints?**
>
> A: Yes, broadly speaking, the hint can be viewed as the initial step of the overall thinking process. However, one of our core innovations is to decouple the verifiable part (hint) from the non-verifiable reasoning part (think). The hint is designed as a precise extraction from the source text. This enables us to compute rewards using objective, automated metrics such as ROUGE, which helps address the non-verifiability challenge introduced at the beginning of the paper. In contrast, the content of think is more flexible and often involves complex logical transitions. Identifying the segment containing “hint” within a long think sequence is more complicated and subjective, and it may bring us back to the difficulty of reward signal design.
>
> > **Q4: In Fig. 4a the hint reward tops near 0.7, but Fig. 4b’s x-axis goes to 1.0. Is 4a truncated or are the scales/normalizations different? Please clarify.**
>
> A: We would like to clarify this question as follows:
> - Figure 4a reports the **average reward across the batch** at each training step. Because each batch contains samples with different levels of difficulty, the batch average hint reward rarely reaches the theoretical maximum of 1.0. A value around 0.7 already indicates strong average performance across the batch.
> - Figure 4b presents a correlation analysis. The horizontal axis shows the full range ([0,1.0]) of **possible hint reward scores for a single sample**. Each point represents "the mean final accuracy of all samples with hint reward equal to x". We use the full range on the horizontal axis to display the complete trend of how increasing hint reward relates to accuracy.

---

> ### Author Response · Authors · 2025-11-26
> **Additional Experiments**
>
> Over the past few days, we have performed additional experiments on hyperparameter analysis in hint reward (Weaknesses 1&2) and an alternative hint structure (Question 3). We hope these results will further address your concerns.
>
> > **W1. The design for the hint reward is rather ad-hoc....**
>
> > **W2. There are many hyperparameters for the reward functions in this proposed system, such as the weights in hint reward,**
>
> To further address your concern regarding the choice of hyperparameters, we have implemented a large-scale experiment on Qwen-32B to analyze the sensitivity of our model to the hyperparameters within the Hint Reward. Specifically, we focused on:
> -  $α$: The weight that balances semantic similarity ($Sim_{cos}$) and lexical overlap ($S_{ROUGE}$) in Equation (1).
> -  $β$: The weight that balances ROUGE-1 and ROUGE-L in Equation (2).
>
> We varied one hyperparameter at a time while keeping the other hyperparameter fixed at our paper's original setting of 0.5. The results (verified by gpt-4o) are presented in the table below:
>
> | Hyperparameter Setting | SBK | CM | SCK | RCB | TA | Avg |
> | :--- | :---: | :---: | :---: | :---: | :---: | :---: |
> | **Analysis of $β$ (with $α=0.5$)** | | | | | | |
> | $β=0.1$ | 0.841 | 0.875 | 0.772 | 0.780 | 0.597 | 0.773 |
> | $β=0.3$ | 0.876 | 0.892 | 0.762 | 0.854 | 0.627 | 0.797 |
> | $β=0.5$ (Original) | 0.867 | 0.800 | 0.812 | 0.839 | 0.672 | 0.798 |
> | $β=0.7$ | 0.805 | 0.808 | 0.752 | 0.902 | 0.746 | **0.802** |
> | $β=0.9$ | 0.867 | 0.817 | 0.713 | 0.756 | 0.687 | 0.768 |
> | **Analysis of $α$ (with $β=0.5$)** | | | | | | |
> | $α=0.1$ | 0.867 | 0.825 | 0.693 | 0.854 | 0.597 | 0.767 |
> | $α=0.3$ | 0.885 | 0.808 | 0.802 | 0.793 | 0.657 | 0.789 |
> | $α=0.5$ (Original) | 0.867 | 0.800 | 0.812 | 0.839 | 0.672 | **0.798** |
> | $α=0.7$ | 0.876 | 0.817 | 0.772 | 0.780 | 0.642 | 0.777 |
> | $α=0.9$ | 0.850 | 0.808 | 0.752 | 0.829 | 0.672 | 0.782 |
>
> From these results, we can draw the following conclusions: 1).  The choice of $α=0.5$ is empirically optimal, the model achieves the highest average score at 0.798 when $α$ is set to 0.5. This suggests that an equal balance between semantic similarity and lexical overlap is the most effective strategy for the Hint Reward. Furthermore, the performance remains relatively stable for $α$ values between 0.3 and 0.9, indicating the model's robustness to this hyperparameter. 2.)  For hyperparameter $β$, its best performance is observed at $0.7$ with avg score of 0.802. Our original setting of $β=0.5$ (0.798) also yields the second highest result.
>
> In summary, these additional experiments demonstrate the robustness of the hyperparameters within our hint reward mechanism. Specifically, the combination of α=0.5 and β=0.7 achieve the best overall performance. We sincerely thank the reviewer for this insightful suggestion, which helped us identify a configuration that further improves our model’s performance. We have updated the manuscript accordingly. The detailed results are reported in Table 3, with the corresponding analysis and trend plots in section 3.5.
>
>
> > **Q3: Can the hint part be viewed as part of the thinking trace? Is it possible to directly incentivize the thinking traces to include these hints?**
>
> Following your suggestion, we implemented a new experiment,  by removing the explicit  `<hint> ` structure and instead incentivized the model to include the hint content directly within the  `<think> ` content. The model receives a binary reward: hint_reward = 1 if the ground-truth hint text is present in its generated thinking content, and 0 otherwise. The results of this new experiment are presented as follows:
>
> | Model | SBK | CM | SCK | RCB | TA | Avg |
> | :--- | :---: | :---: | :---: | :---: | :---: | :---: |
> | Qwen2.5-32B-Instruct | 0.8318 | 0.7458 | 0.7277 | 0.6341 | 0.5373 | 0.6953 |
> | Qwen2.5-32B-GRPO-hint-within-think | 0.8362 | 0.825| 0.7970 | 0.8292 | 0.4328 | 0.7440 |
> | Qwen2.5-32B-GRPO | 0.8805 | 0.8333 | 0.8762 | 0.8573 | 0.6865 | 0.8268 |
>
> The results show that although the performance of Qwen2.5-32B-SFT-hint-within-think is better than Qwen2.5-32B-Instruct, its performance still has a significant gap compared to that of Qwen2.5-32B-GRPO. This demonstrates that while hint information is indeed beneficial for improving role-playing performance, the method of utilization significantly impacts the level of improvement. Our approach, which decouples the hint for explicit verification, is more effective than simply incentivizing its inclusion within the thinking process.

---

### Author Response · Authors · 2025-12-02
**Summary of Contributions and Responses**

Dear Reviewers, ACs, SACs and PCs,

We sincerely thank you all for your time, constructive feedback and selfless dedication. We have carefully considered all comments and conducted additional experiments to address the concerns of reviewers. Below is a summary of our paper's contributions, and our efforts to address the weaknesses and questions raised during the review process.

**1. Summary of Contributions**

Our paper addresses the **"non-verifiability"** challenge in applying Reinforcement Learning to Role-Playing Conversational Agents. Our core contributions are:
- We propose VeriRole, a framework that decouples the reasoning process into a "Hint-Think-Reply" structure. By extracting verifiable factual cues (hints) from the context, we can apply objective rewards to guide the open-ended generation.
- We build two specialized datasets for role-playing scenarios, and employ a verifiable reward design to enhance role-awareness and factual consistency
- We conduct extensive experiments across multiple baselines and benchmarks, and we present detailed ablation studies and an in-depth analysis of the hint mechanism. The results demonstrate the effectiveness of our proposed VeriRole.

**2. Summary of Responses to Reviewers' Comments**

**2.1. To Reviewer vfNc:**
> We paid special attention to the concerns raised by Reviewer vfNc. Although the score cannot be changed at this stage, we believe our additional experiments and clarifications have fully addressed these critical points:
- **Hyperparameter Sensitivity (W1&2)**: The reviewer expressed concern that the hint reward design seemed ad-hoc and relied on many hyperparameters. In response, we conducted a large-scale sensitivity analysis on the weights  $α$ and $β$. The results in Table 3 demonstrate that our original setting is close to optimal and the model performance is robust within a reasonable range of these parameters.
- **Generalizability of method (W3)**: We clarified that our core idea of "extracting verifiable cues" is generalizable to tasks like summarization, and the Situation Puzzle results further prove its potential for complex reasoning.
- **Necessity of Hint Reward (W4)**: Following the reviewer's suggestion, we added an ablation study that completely removes the Hint Reward. The results show a significant performance drop. This empirically proves that the hint mechanism is not redundant.
- **Incentivizing Hint (Q1)**: We clarified the mechanism that forces the model to use the generated hint. The prompt explicitly requires the model to outline logic based on hints.
- **Qwen3-think (Q2)**: We clarified the generation format of Qwen3-Think baseline.
- **Hint vs Thinking trace (Q3)**: We conducted a new experiment of rewarding hints directly within the thinking trace. The lower performance confirms that our explicit, decoupled hint structure is more effective.
- **Clarification on Figure 4 (Q4)**: We resolved the confusion regarding the axes in Figure 4. We explained that Figure 4a reports the batch average of hint reward, while Figure 4b shows the full range of possible hint score.

**2.2. To Reviewer HVvs:**
- **Reward Balancing**: We clarified that we balance the Hint, Accuracy, and Format rewards by setting their maximum scales (1.0, 1.0, and 0.6 respectively). Figure 4a shows the effectiveness of our setting.
- **Hint Quality**: We detailed our multi-stage filtering pipeline which ensures the ground-truth hints are precise and objective.

**2.3. To Reviewer JWuD:**
- **Cost and Transferability**: We applied our pipeline to a separate RoleLLM dataset, and found that the average time cost is 2.5 seconds per sample, with a valid hint extraction rate of 66.7%.
- **Creativity Trade-off**: To solve reviewer's concern of sacrificing creativity for accuracy, we added additional analysis showing that our model with accuracy reward improves on the "Engagingness" metric in CharacterEval.
- **Ablation Study**: We clarified that the "No-Accuracy-Reward" setting removes the accuracy signal entirely, which hurts performance on factual tasks.

**2.4. To Reviewer ccyV:**
- **RL vs. SFT**: To verify the necessary of RL, we added a new experiment comparing our method against a mixed-objective SFT baseline (training on hint+reply).  The result shows our GRPO significantly outperformed the SFT baseline.
- **$H_{gt}$ Scalability**: We explained that our method is robust even when hints are not applicable (e.g. Chit-Chat) by training the model to generate empty hints, ensuring scalability across different dialogue types.
- **$R_{acc}$ on Creative Tasks**: We clarified that the extracted hint related to creative objectives is effective enough to guide the following generation,  without the corresponding $R_{acc}$.

---

### Meta-Review · Area_Chair_dEZk · 2026-01-03

**Summary:**

This paper addresses a key challenge in applying reinforcement learning to role-playing conversational agents: role-awareness is difficult to reward because it is often non-verifiable. The authors propose VeriRole, a framework that decomposes generation into a Hint–Think–Reply structure, where verifiable "hints" are first extracted from the context and used to define an objective reward signal for guiding open-ended responses. The AC considers this problem to be of interest to broader settings beyond role-playing conversational agents, where reward design is similarly challenging due to non-verifiability.

The paper presents extensive experiments and ablations across multiple models and benchmarks. Reviewers agree that the problem is important and that the proposed solution is well-motivated and clearly presented. Empirically, VeriRole achieves strong improvements on RAIDEN and consistent gains on CharacterEval, supporting the effectiveness of using verifiable intermediate signals to optimize role-aware behavior.

**Reviewer Concerns:**

From the AC’s reading, the authors have responded effectively to the reviewers’ main concerns, including those related to ablations, reward design, and empirical validation. However, it is difficult to predict whether these responses would be sufficient for reviewers to substantially revise their scores.

One remaining concern noted by the AC is the dependence on LLM-as-judge evaluation and the potential bias this introduces. The evaluation relies heavily on GPT-4o/Claude-based judgments. While the authors take steps to mitigate this through multiple benchmarks and validation strategies, this limitation remains inherent to the evaluation setup. At present, the AC does not have a clear alternative suggestion that would more effectively address this concern within the scope of the current work.

**Reviewer Scores:**

Based on the rebuttal and additional experimental results, the AC believes that the overall reviewer sentiment would converge toward a consensus that this paper is of interest to the community and merits acceptance. The authors have substantively addressed the major technical concerns raised during review, particularly those regarding the necessity of the hint mechanism and the design of the reward function.

---

### Decision · Program_Chairs · 2026-01-26

Accept (Poster)